# Globally optimal score-based learning of directed acyclic graphs in high-dimensions

**Bryon Aragam**[1]     **Arash A. Amini**[2]     **Qing Zhou**[2]
[1]University of Chicago    bryon@chicagobooth.edu
[2]University of California, Los Angeles    {aaamini,zhou}@stat.ucla.edu

## Abstract

We prove that $\Omega(s \log p)$ samples suffice to learn a sparse Gaussian directed acyclic graph (DAG) from data, where $s$ is the maximum Markov blanket size. This improves upon recent results that require $\Omega(s^4 \log p)$ samples in the equal variance case. To prove this, we analyze a popular score-based estimator that has been the subject of extensive empirical inquiry in recent years and is known to achieve state-of-the-art results. Furthermore, the approach we study does not require strong assumptions such as faithfulness that existing theory for score-based learning crucially relies on. The resulting estimator is based around a difficult nonconvex optimization problem, and its analysis may be of independent interest given recent interest in nonconvex optimization in machine learning. Our analysis overcomes the drawbacks of existing theoretical analyses, which either fail to guarantee structure consistency in high-dimensions (i.e. learning the correct graph with high probability), or rely on restrictive assumptions. In contrast, we give explicit finite-sample bounds that are valid in the important $p \gg n$ regime.

## 1  Introduction

With the growing importance of explainability and interpretability in modern machine learning [11, 64, 65], graphical models continue to play an important role in applications including genomics [72], health care [41], and finance [50] owing to their natural interpretability and simplicity. For this reason, rigorous theoretical understanding of graphical models is an important challenge in modern machine learning. Although estimating undirected graphical models can be formulated as a convex program, DAG models cannot be [15], which has limited our understanding of their finite-sample properties. Despite impressive progress in our understanding of nonconvex models across a spectrum of problems including dictionary learning [58], tensor decomposition [16, 18], deep neural networks [13, 14], and regression [36, 37], learning DAGs remains an important problem with many open questions, particularly in the high-dimensional ($p \gg n$) setting.

Among the many strategies for learning DAGs from data, score-based learning is a classical approach that is popular in practice. While much is known about greedy search algorithms [6, 40], much less is known regarding the statistical properties of methods that find a global minimizer of a score function. One of the advantages of the latter approach is a potential relaxation of assumptions such as faithfulness [61]. In this paper, we prove that a score-based method requires only $O(s \log p)$ samples, where $s$ is the maximum Markov blanket size, at the cost of being difficult to compute since it requires solving a nonconvex, NP-hard optimization problem. This is a well-known drawback of score-based methods, although recent work has demonstrated that approximate methods can outperform state-of-the-art methods [1, 25, 70], and even come close to finding the global minimum in practice [77].

More specifically, we characterize the finite-sample, high-dimensional behaviour of the following score-based DAG estimator, formulated as the solution of a constrained, nonsmooth, nonconvex

optimization problem:

$$\widehat{B} \in \underset{B \in \mathbb{D}}{\arg\min} \, Q(B), \quad Q(B) = \frac{1}{2n} \left\| \mathbf{X} - \mathbf{X}B \right\|_F^2 + \rho_\lambda(B), \tag{1}$$

where $\mathbb{D}$ is the set of $p \times p$ matrices representing the weighted adjacency matrix of a DAG, $\mathbf{X} \in \mathbb{R}^{n \times p}$ is the data, and $\rho_\lambda$ is a suitably chosen regularizer (Section 2.3). In the literature on learning DAGs, $Q$ is called a *score function*. This estimator has been the subject of extensive empirical inquiry [e.g. 1, 26, 51, 54, 68, 77], and outperforms classical approaches such as the PC algorithm [57] and greedy equivalence search [GES, 6] on high-dimensional data. Moreover, although computation of $\widehat{B}$ is NP-hard [7], it can be computed exactly using dynamic programming [43, 44, 55, 56] and mixed integer programs [9, 10], and approximate algorithms for computing this estimator scale to modern problem sizes with tens of thousands of variables [1, 3].

**Contributions**    In this paper we provide a comprehensive portrait of the behaviour of $\widehat{B}$, providing much needed justification—and caution—for its use in applications. Specifically, our main contributions are as follows:

1. We provide explicit, finite-sample structure recovery guarantees for the score-based estimator (1) that are valid when $p \gg n$. This is in contrast to recent work on score-based methods that either studies asymptotic properties of specific algorithms under faithfulness [40], or does not prove exact structure recovery [35, 62].

2. We develop a new proof technique in order to simplify the analysis of score-based estimators, based on a novel lattice construction and a reduction to neighbourhood regression. This construction allows us to provide *uniform* control over the superexponential family of neighbourhood regression problems that define (1), a result that is potentially interesting in its own right.

3. We use this construction to prove an $\Omega(s \log p)$ sample complexity under which $\widehat{B}$ recovers the true DAG with high probability, which improves upon existing results. We also generalize existing results on estimating identifiable DAGs with equal error variances to what we call *minimum-trace* DAGs.

4. We discuss the more general, nonidentifiable case. In this setting, there is no "truth" to approximate, however, we show that $\widehat{B}$ still estimates a sufficiently sparse representative of the underlying distribution.

We anticipate these results will be of interest not only to the graphical modeling community, but also to the broader machine learning community in the way it analyzes a difficult nonconvex optimization problem head on.

**Previous work**    It was recently shown that it is possible to learn DAGs in high-dimensions [21–23, 67]. These papers prove a lower bound of $\Omega(k \log p)$ on the sample complexity where $k$ is the maximum number of parents in the true DAG, and provide a polynomial-time algorithm that requires $O(s^4 \log p)$ samples to recover this DAG. These papers are based on a new approach—distinct from traditional score-based or constraint-based learning—that uses second-order information to find a node ordering. Once this ordering is found, estimation is straightforward. Earlier work on the linear non-Gaussian case uses independent component analysis to identify the true DAG model [52, 53] but requires $n > p$ and as such is not high-dimensional.

Perhaps surprisingly, despite score-based methods being very popular in practice, none of these papers consider score-based methods. Asymptotically, consistency of the score-based GES algorithm is well-known [6, 40], however, to the best of our knowledge finite-sample complexity results are not available for GES. Furthermore, these results assume strong faithfulness, which—as the name suggests—is an even stronger version of faithfulness that is known to be very stringent and may not hold in practice [34, 61]. By assuming faithfulness, the Markov equivalence class—and hence CPDAG—of a distribution becomes identified, which greatly simplifies the theoretical analysis. Only a few recent papers have studied finite-sample properties of score-based estimators: van de Geer and Bühlmann [62] establish $\ell_2$-consistency of a restricted $\ell_0$-regularized MLE, Loh and Bühlmann [35] analyze the empirical score of DAGs that are consistent with an estimated moral graph, and Yuan et al. [71] analyze a constrained MLE. Unfortunately, the practical implications of these interesting theoretical results have been limited by certain aspects of their analysis. Although van de Geer and

Bühlmann [62] and Yuan et al. [71] avoid the faithfulness assumption, their structure consistency results require $p \leq n$ and thus do not provide a direct theory for the high-dimensional structure learning problem. Loh and Bühlmann [35] do not consider the problem of structure recovery, and one of our contributions is to show that by properly regularizing the score in high-dimensions, structure recovery is possible when $p \gg n$.

Perhaps surprisingly, proving consistency for the global minimizer of (1) turns out to be a unique challenge: Despite a growing literature on theory for nonconvex problems [5, 8, 13, 14, 16, 18, 19, 28–30, 33, 38, 58, 60], existing techniques from the graphical modeling literature fail to capture the essence of the program (1). Classical arguments such as the basic inequality can be used to prove $\ell_2$-rates of convergence as in [63], but translating these rates into structure learning (e.g. by thresholding) requires $n = \Omega(p)$. By assuming strong faithfulness, one can simplify the problem substantially by reducing it to a constraint-based method as in [40]. The latter work in particular sidesteps all of the difficulties in analyzing the nonconvex program (1), which constitute arguably some of the most interesting theoretical aspects of this problem. More discussion on these points can be found in Section 6.

## 2 Background

Our approach is based on the structural equation model (SEM) interpretation of Gaussian DAGs. Suppose $X = (X_1, \ldots, X_p)$ is a random vector satisfying

$$X = \widetilde{B}^T X + \widetilde{\varepsilon}, \quad \widetilde{\varepsilon} \sim \mathcal{N}_p(0, \widetilde{\Omega}), \tag{2}$$

where $\widetilde{B} \in \mathbb{D}$ and $\widetilde{\Omega}$ is a $p \times p$ positive diagonal matrix of variances. One can interpret $\widetilde{B}$ as the weighted adjacency matrix of a graph. Given an $n \times p$ random matrix $\mathbf{X}$ whose rows are i.i.d. drawn according to the model (2), we define a penalized least-squares (PLS) score function by (1). It follows from (2) that $X \sim \mathcal{N}_p(0, \Sigma(\widetilde{B}, \widetilde{\Omega}))$, where

$$\Sigma(\widetilde{B}, \widetilde{\Omega}) := (I - \widetilde{B})^{-T} \widetilde{\Omega} (I - \widetilde{B})^{-1}. \tag{3}$$

We will assume that $\Sigma \succ 0$, and moreover that $r_{\min}(\Sigma) \asymp r_{\max}(\Sigma) \asymp 1$, i.e. the eigenvalues of $\Sigma$ are bounded away from 0 and $\infty$. This is purely to simplify the theorem statements; explicit constants depending on $\Sigma$ and its eigenvalues can be found in the supplement.

**Notation** We write $a \gtrsim b$ (resp. $a \lesssim b$) to mean that $a \geq C \cdot b$ (resp. $a \leq C \cdot b$) for some constant $C > 0$. In all cases, exact values for these constants can be found in the supplement.

### 2.1 Identifiability

The map $(\widetilde{B}, \widetilde{\Omega}) \mapsto \Sigma(\widetilde{B}, \widetilde{\Omega})$ is not one-to-one, i.e. without further assumptions the model (2) is nonidentifiable. Recent work [12, 21, 62] assumes *equivariance*, i.e. $\Sigma = \Sigma(\widetilde{B}, \widetilde{\omega}^2 I)$ for some $\widetilde{\omega}^2 > 0$, which ensures that $\widetilde{B}$ is identifiable [47]. We generalize this condition as follows: Let $\mathbb{R}_+^p$ denote the space of $p \times p$ positive diagonal matrices and define the *equivalence class* of $\Sigma$ by

$$\mathfrak{D}(\Sigma) = \left\{ (\widetilde{B}, \widetilde{\Omega}) \in \mathbb{D} \times \mathbb{R}_+^p : \Sigma = \Sigma(\widetilde{B}, \widetilde{\Omega}) \right\}, \tag{4}$$

and call $\widetilde{B}_{\min}$ a *minimum-trace DAG* if $(\widetilde{B}_{\min}, \widetilde{\Omega}_{\min}) \in \arg\min\{\operatorname{tr} \widetilde{\Omega} : (\widetilde{B}, \widetilde{\Omega}) \in \mathfrak{D}(\Sigma)\}$. In other words, $\widetilde{B}_{\min}$ minimizes the total conditional variance amongst all of the DAGs that represent $\Sigma$. We will sometimes abuse notation by writing $\widetilde{B} \in \mathfrak{D}(\Sigma)$ or $\widetilde{\Omega} \in \mathfrak{D}(\Sigma)$ for short. The following lemma connects equivariant DAGs to minimum-trace DAGs:

**Lemma 2.1.** *Suppose $\Sigma$ is given and $\Sigma = \Sigma(\widetilde{B}, \widetilde{\omega}^2 I)$ for some $\widetilde{\omega}^2 > 0$. Then $\widetilde{B}$ is the unique minimum-trace DAG in $\mathfrak{D}(\Sigma)$.*

In general, minimum-trace DAGs are not unique, so this lemma shows that the concept of minimum-trace provides a convenient generalization of known identifiability results for equivariant DAGs.

Beyond their connection with equivariance DAGs, it is important to address *why* minimum-trace DAGs should be of interest in the sequel. As discussed previously, despite a lack of theoretical

justification, the estimator $\widehat{B}$ is popular in practice [e.g. 1, 26, 51, 54, 68, 77]. Our motivation is to answer fundamental questions such as *does $\widehat{B}$ converge*, and if so, *to what*? We note that even the former question is surprisingly tricky; see Section 4. The results presented in this paper will show that not only does $\widehat{B}$ converge, we can pinpoint what it converges to, namely a minimum-trace DAG. The importance of this result lies not in the fact that we might be interested in minimum-trace DAGs, but perhaps that we might not be: Whether or not one would be interested in a minimum-trace (or equivariance) DAG depends on the application.

## 2.2 Superstructures

In addition to (1), we will also study a restricted version of $\widehat{B}$ defined as follows: Given an undirected graph $G = (V, E)$, define $\mathbb{D}_G = \{B \in \mathbb{D} : B \subset G\}$, i.e. the subset of $\mathbb{D}$ that are subgraphs of $G$, and

$$\widehat{B}(G) \in \underset{B \in \mathbb{D}_G}{\arg\min} Q(B), \tag{5}$$

where $Q(B)$ is defined as in (1). The graph $G$ is called a *superstructure*, and reduces both the computational and statistical complexity of score-based methods [42, 46]. We recall here also the *moral graph* $\mathfrak{m}(B)$ of a DAG $B$, defined as the undirected graph that results from ignoring edge orientation in $B$ and adding an undirected edge between the parents of each node in $B$. Clearly, $\mathfrak{m}(B)$ is a superstructure of $B$.

## 2.3 Regularizer

Traditionally, score functions use $\ell_0$-regularization, i.e. $\rho_\lambda(B) = \lambda^2 \sum_{i,j} 1(\beta_{ij} \neq 0)$ [6, 20, 62]. This penalty leads to good theoretical properties but is difficult to optimize due to its combinatorial nature. For this reason, we consider the $\ell_1$-regularizer, $\rho_\lambda(B) = \lambda \sum_{i,j} |\beta_{ij}|$, which is a convex surrogate of the $\ell_0$-regularizer that is easier to optimize [77], as well as the minimax concave penalty (MCP) [73], which is a continuous, nonconvex interpolant between $\ell_0$ and $\ell_1$ regularization. Although easier to compute with, $\ell_1$ regularization is known to require strong incoherence conditions for consistent variable selection [39, 66, 76], whereas the MCP does not require these conditions. More details can be found in Appendix A.2 of the supplement.

The following condition formalizes the assumptions we place on $\rho_\lambda$. Let $N_j(G)$ denote the neighbourhood of $X_j$ in $G$, i.e. the set of all vertices adjacent to $X_j$.

**Condition 2.1** (Regularizer). *The regularizer $\rho_\lambda$ is either $\ell_1$ or the MCP. If $\ell_1$ regularization is used, then additionally assume that $\zeta(G) < 1$, where*

$$\zeta(G) := \sup_{1 \leq j \leq p} \sup_{S \subset N_j(G)} \|\Sigma_{S^c S}(\Sigma_{SS})^{-1}\|_{1,\infty}. \tag{6}$$

Here, $\|A\|_{1,\infty} = \max_i \sum_j |a_{ij}|$. Crucially, if $\rho_\lambda$ is the MCP, then we are left with a continuous optimization problem *without* requiring any incoherence conditions.

# 3 The identifiable case: Recovery of minimum-trace DAGs

We begin with the identifiable case, i.e. $\widetilde{B}_{\min}$ is unique.

## 3.1 Assumptions

Given a minimum-trace DAG $\widetilde{B}_{\min}$, define for $\eta > 0$

$$\chi(\eta) := \inf_{\substack{\widetilde{\Omega} \neq \widetilde{\Omega}_{\min} \\ \widetilde{\Omega} \in \mathfrak{D}(\Sigma)}} \left[ (1 - \eta) \operatorname{tr} \widetilde{\Omega} - (1 + \eta) \operatorname{tr} \widetilde{\Omega}_{\min} - \rho_\lambda(\widetilde{B}_{\min}) \right]. \tag{7}$$

Given a superstructure $G$, let $s = s(G)$ denote the maximum degree of $G$, and define

$$\gamma_1 = \gamma_1(G) := 4\sqrt{\frac{s \log[3ep/s] + \log p}{n}}, \tag{8}$$

$$\gamma_2 = \gamma_2(G) := \left(1 + 3\sqrt{2}\sqrt{\frac{s \log(ep/s)}{n}}\right)^2. \tag{9}$$

**Condition 3.1** (Identifiability). $\Sigma \succ 0$, and

(a) There exists a unique minimum-trace DAG $\widetilde{B}_{\min} \in \mathfrak{D}(\Sigma)$;

(b) $\gamma_1(G) \leq 1$ and $\chi(\eta) > 0$, where $\eta := \gamma_1[1 + 6\kappa(\Sigma; s)\gamma_2]$ and $\kappa(\Sigma; s)$ is a constant that depends on $\Sigma$ and $s$.

See (47) in the supplement for an exact expression of $\kappa(\Sigma; s)$, which is roughly the maximum condition number of the principal submatrices of $\Sigma$ of size $O(s)$. Condition 3.1(a) is an identifiability condition on $\widetilde{B}_{\min}$, and Condition 3.1(b) is needed to recover $\widetilde{B}_{\min}$ from finite samples. By Lemma 2.1, Condition 3.1(a) is strictly weaker than equivariance. Under this condition, we can speak of "the" minimum-trace DAG, which will be denoted in the sequel by $(\widetilde{B}_{\min}, \widetilde{\Omega}_{\min})$. Condition 3.1(b) is closely related to gap conditions that have appeared previously [35, 62], and is discussed in detail in Section 3.2.

## 3.2 First main result: Identifiable DAGs

For any $A \in \mathbb{R}^{p \times p}$, let $\tau_*(A) := \min\{|a_{ij}| : a_{ij} \neq 0\}$. The quantity $\tau_*(\widetilde{B}_{\min})$ measures the smallest nonzero weight in $\widetilde{B}_{\min}$, which is a measure of the signal strength in the problem.

**Theorem 3.1.** *Suppose that Conditions 2.1 and 3.1 hold and that $\widetilde{B}_{\min} \subset G$. If $n \gtrsim s \log p$, $\lambda \gtrsim \sqrt{\log p / n}$, and $\tau_*(\widetilde{B}_{\min}) \gtrsim \lambda$, then*

$$\mathrm{supp}(\widehat{B}(G)) = \mathrm{supp}(\widetilde{B}_{\min})$$

*with probability $1 - O(e^{-k \log p})$, where $k$ is the maximum in-degree of $\widetilde{B}_{\min}$.*

In fact, even if Condition 3.1(a) fails—i.e. $\widetilde{B}_{\min}$ is not identifiable—the conclusions of Theorem 3.1 continue to hold for *some* minimum-trace DAG. In the next section, we consider the nonidentifiable case in even greater detail (see Theorem 4.1).

The previous theorem assumes that a *consistent* superstructure $G$ is known, i.e. that $\widetilde{B}_{\min} \subset G$. A standard approach is to define $G$ by the support of a consistent estimate of the precision matrix $\Gamma = \Sigma^{-1}$. The following assumption encodes the minimal requirement we need on $\Sigma$ and $\widetilde{B}_{\min}$:

**Condition 3.2** (Superstructure). *If $(i, j)$ is an edge in $\mathfrak{m}(\widetilde{B}_{\min})$, then $\Gamma_{ij} \neq 0$.*

The results in Loh and Bühlmann [35] show that as long as the entries of $\widetilde{B}_{\min}$ are drawn from a continuous distribution, Condition 3.2 is satisfied except on a set of measure zero. For details, see Theorem 2 and Assumption 1 therein. Under Condition 3.2, it suffices to use a consistent estimate of the support of $\Gamma$, which can be estimated using known results [39]. Let $\widehat{\Gamma}$ denote such an estimate and with some abuse of notation, denote the resulting DAG estimator by $\widehat{B}(\widehat{\Gamma})$.

**Corollary 3.1.** *Suppose that Conditions 2.1, 3.1, and 3.2 hold. If $n \gtrsim s \log p$, $\lambda \gtrsim \sqrt{\log p / n}$, $\tau_*(\Gamma) \gtrsim \lambda$, and $\tau_*(\widetilde{B}_{\min}) \gtrsim \lambda$, then*

$$\mathrm{supp}(\widehat{B}(\widehat{\Gamma})) = \mathrm{supp}(\widetilde{B}_{\min})$$

*with probability $1 - O(e^{-k \log p})$.*

Corollary 3.1 implies that there is a score-based estimator with sample complexity $\Omega(s \log p)$. In contrast to Ghoshal and Honorio [21], who require an element-wise consistent estimate of $\Gamma$ (i.e. in $\ell_\infty$-norm), our result only requires the support of $\Gamma$. The former approach leads to a $\Omega(s^4 \log p)$ sample complexity, whereas our approach requires only $\Omega(s \log p)$ samples. Both of these results are a significant improvement over existing results on score-based methods, e.g. Theorem 5.1 in [62], which requires $p \lesssim n / \log n$ and hence $n \gtrsim p$.

**Faithfulness and the beta-min condition** Theorem 3.1 does not require the faithfulness assumption, which is a standard assumption in the literature on learning DAGs for both score-based [6, 40] and constraint-based methods [31], and is known to be very strong in practice [34, 61]. Assuming

faithfulness, the Markov equivalence class becomes identified, which simplifies the problem by restricting the number of equivalent DAGs that must be controlled. Recent work has also relaxed this assumption [21, 23, 45, 62], however, to the best of our knowledge, our result is the first such result for score-based estimators in high-dimensions. Instead, we require a beta-min condition on the true DAG $\widetilde{B}_{\min}$, which is typical in the statistical literature on model selection.

**Gap condition**   Condition 3.1(b) imposes an implicit assumption on the degree of $G$ through the requirement $\gamma_1(G) \leq 1$ which roughly translates to $s \log(p/s) + \log p \lesssim n$. The assumption on $\chi(\eta)$, on the other hand, is a type of identifiability condition on $\widetilde{B}_{\min}$. Whereas Condition 3.1(a) requires $\widetilde{B}_{\min}$ to be identifiable in the infinite sample limit, Condition 3.1(b) requires that there is a "gap" on the order $\sqrt{s \log p / n}$ between the expected loss of $\widetilde{B}_{\min}$ and the expected loss of any other DAG in $\mathfrak{D}(\Sigma)$. To see this, note that $\mathbb{E}\|\mathbf{X} - \mathbf{X}\widetilde{B}\|_F^2 / n = \operatorname{tr}\widetilde{\Omega}$ for any $(\widetilde{B}, \widetilde{\Omega}) \in \mathfrak{D}(\Sigma)$ and define

$$\operatorname{gap}(\Sigma) := \inf \left\{ \operatorname{tr}\widetilde{\Omega} - \operatorname{tr}\widetilde{\Omega}_{\min} : \widetilde{\Omega} \neq \widetilde{\Omega}_{\min}, \ \widetilde{\Omega} \in \mathfrak{D}(\Sigma) \right\}. \tag{10}$$

When $\rho_\lambda$ is the MCP and $\lambda \gtrsim \eta$, a straightforward calculation shows that the following two conditions are sufficient to guarantee Condition 3.1(b), in addition to $\gamma_1 \leq 1$: There exists $a \geq 0$ such that

$$\operatorname{gap}(\Sigma) \gtrsim \left[ \frac{s \log(p/s) + \log p}{n} \right]^a p, \tag{11}$$

$$\|\widetilde{B}_{\min}\|_0 \lesssim \left[ \frac{n}{s \log(p/s) + \log p} \right]^{1 - \frac{a}{2}} p. \tag{12}$$

Thus, Condition 3.1(b) allows one to trade off the size of the "gap" in (11) with a sparsity condition (12) on $\widetilde{B}_{\min}$. For example, taking $a \in (0, 2)$ and $s \log(p/s) + \log p \ll n$ allows $\operatorname{gap}(\Sigma) = o(p)$ while simultaneously tolerating an average degree $\|\widetilde{B}_{\min}\|_0 / p$ that grows without bound (cf. (12)). Since the problem considered here is at least as hard as $p$ separate regression problems, this scaling in terms of $p$ is expected. Similar conditions with a similar scaling have appeared in previous work [35, 62].

## 4   The general case: Recovery of sparse representations

In the previous section, we leveraged strong prior information—namely identifiability and a consistent superstructure—in order to analyze the sample complexity of learning a minimum-trace DAG. In practice, such prior information may not be available, and in general it is well-known that Gaussian DAGs are not identifiable [2, 62]. The estimator (1), of course, is well-defined whether or not Condition 3.1 holds, and in practice, one typically computes $\widehat{B}$ and "hopes for the best". Is it possible to say more in the general setting? Surprisingly, even if there is no DAG $\widetilde{B} \in \mathfrak{D}(\Sigma)$ that is identifiable, we can still provide guarantees. The idea is to first show that $\widehat{B}$ converges to *some* DAG $\widetilde{B} \in \mathfrak{D}(\Sigma)$, and then show that $\widetilde{B}$ is well-behaved compared to other representative DAGs in $\mathfrak{D}(\Sigma)$. Specifically, we will show that $\widetilde{B}$ is roughly as sparse as a minimum-trace DAG.

### 4.1   Assumptions

*Definition* 4.1.   Let $\widetilde{\beta}_j$ denote the $j$th column of $\widetilde{B} \in \mathbb{D}$. For any $\Sigma$, let

$$d(\mathfrak{D}(\Sigma)) := \sup_{\widetilde{B} \in \mathfrak{D}(\Sigma)} \|\widetilde{B}\|_0, \quad \tau_*(\mathfrak{D}(\Sigma)) := \inf_{\widetilde{B} \in \mathfrak{D}(\Sigma)} \tau_*(\widetilde{B}). \tag{13}$$

We will write $d = d(\mathfrak{D}(\Sigma))$ to simplify the notation in the sequel.

**Condition 4.1** (Minimum-trace DAG).   $\Sigma \succ 0$, *and there is a minimum-trace DAG $\widetilde{B}_{\min}$ such that*

$$\frac{\rho_\lambda(\widetilde{B}_{\min})}{\operatorname{tr}\widetilde{\Omega}_{\min}} \geq a_2 \sqrt{\frac{(d+1)\log p}{n}} \quad \textit{for some } a_2 > 0.$$

Condition 4.1 can be interpreted as putting a soft lower bound on the weights in $\widetilde{B}_{\min}$, as measured by the regularizer $\rho_\lambda$ and $\widetilde{\Omega}_{\min}$. For comparison, recall that the usual beta-min condition in regression is $\min_j |\beta_j| \gtrsim \sigma \sqrt{\log p / n}$.

## 4.2 Second main result: The nonidentifiable case

Our second result shows that even in the absence of identifiability assumptions, we can still guarantee that $\widehat{B}$ recovers the support of a DAG $\widetilde{B} \in \mathfrak{D}(\Sigma)$, and that $\widetilde{B}$ must also be sparse. In fact, we note that even without the sparsity conclusion, it is not obvious (and indeed nontrivial to show) that $\widehat{B}$ approaches any particular member of $\mathfrak{D}(\Sigma)$.

**Theorem 4.1.** *Suppose that Conditions 2.1 (with $G$ the complete graph in the case of $\ell_1$) and 4.1 hold. If $n \gtrsim d \log p$, $\lambda \gtrsim \sqrt{(d+1) \log p / n}$, and $\tau_*(\mathfrak{D}(\Sigma)) \gtrsim \lambda$ then there exists $\widetilde{B} \in \mathfrak{D}(\Sigma)$ and a minimum-trace DAG $\widetilde{B}_{\min} \in \mathfrak{D}(\Sigma)$ such that*

$$\mathrm{supp}(\widehat{B}) = \mathrm{supp}(\widetilde{B}) \quad and \quad \rho_\lambda(\widetilde{B}) \lesssim \rho_\lambda(\widehat{B}) \lesssim \rho_\lambda(\widetilde{B}_{\min})$$

*with probability at least $1 - O(e^{-d \log p})$.*

This is similar to the approach taken in van de Geer and Bühlmann [62] with some key differences: 1) Their Theorem 3.1 does *not* establish structure consistency, and 2) Their $\ell_0$-regularized MLE involves a thresholded parameter space that is much more difficult to compute in practice, whereas our estimator (1) is defined over the full parameter space and involves continuous optimization.

In contrast to Theorem 3.1, Theorem 4.1 no longer requires the identifiability condition (Condition 3.1), which is replaced by Condition 4.1 on $\widetilde{B}_{\min}$. The tradeoffs are 1) The estimator $\widehat{B}$ is no longer guaranteed to recover an exact minimum-trace DAG, and 2) The beta-min condition and sample complexity now depend on the sparsity parameter $d$, which may be larger than $s$ and can be large for general covariance matrices. This result also emphasizes the advantages of nonconvex regularization: When $\ell_1$-regularization is used, the incoherence condition (6) is imposed over every neighbourhood, which is a very severe restriction. With the MCP, there are no incoherence assumptions whatsoever.

**Sparsity** A key conclusion in Theorem 4.1 is that $\rho_\lambda(\widetilde{B}) \lesssim \rho_\lambda(\widehat{B}) \lesssim \rho_\lambda(\widetilde{B}_{\min})$: This says that $\widehat{B}$ is consistent with a parsimonious DAG. It is easy to show that this implies $\|\widetilde{B}\|_0 \lesssim \|\widehat{B}\|_0 \lesssim \|\widetilde{B}_{\min}\|_0$ for the MCP regularizer. For the $\ell_1$ penalty, we have $\|\widetilde{B}\|_1 \lesssim \|\widehat{B}\|_1 \lesssim \|\widetilde{B}_{\min}\|_1$, which can be interpreted as a "soft" notion of sparsity.

**Strong faithfulness and the beta-min condition** In contrast to Theorem 3.1, which only requires a beta-min condition on the true DAG $\widetilde{B}_{\min}$, Theorem 4.1 requires a much stronger condition on the smallest weight of any DAG in the equivalence class $\mathfrak{D}(\Sigma)$ (cf. (13)). This is reminiscent of—but not the same as—the *strong faithfulness* condition, which roughly asserts that the minimum partial correlation between any pair of $d$-separated variables in the true DAG is bounded away from zero. We leave it to future work to study this connection more carefully, however, we note here that previous work on this problem [61] has noted the difficulty of establishing such an explicit relationship, and to the best of our knowledge this remains an open problem. Nonetheless, the novelty of Theorem 4.1 is in establishing finite-sample structure recovery without imposing any identifiability requirement, so it is natural to expect that stronger assumptions will be needed.

## 5 Proof outline

Our basic strategy is to reduce the analysis of $\widehat{B}$ to a family of neighbourhood regression problems, using a similar approach as in our preprint [2]. This is similar to undirected models, for which the analysis can be reduced to $p$ different regression problems, namely the regression of $X_j$ onto $X_{-j}$ [39, 69]. Unfortunately, for DAGs, there are $p2^p$ possible regression problems (the regression of $X_j$ onto any subset $S \subset [p]_j$), which quickly become intractable to control uniformly. The manner in which these problems are controlled highlights the main technical difference between the proofs of Theorems 3.1 and 4.1.

To prove Theorem 3.1, we first prove a uniform concentration result for the score $Q(B)$. Specifically, letting $\ell(B) = \|\mathbf{X} - \mathbf{X}B\|_F^2 / (2n)$, we show that the following upper bound holds with high probability over $\mathbb{D}_G$ (Proposition B.7):

$$|\ell(B) - \mathbb{E}\ell(B)| \leq \gamma_1 [1 + 6\kappa(\Sigma; s)\gamma_2] \mathbb{E}\ell(B) \quad \text{for all } B \in \mathbb{D}_G. \tag{14}$$

Based on this result, we show that $\widehat{B}$ has the same topological sort as $\widetilde{B}_{\min}$. This topological sort identifies candidate parent sets for each node $X_j$, and reduces the problem to $p$ regression problems. The main technical device here is uniform score concentration via (14), which is an interesting result in its own right due to its uniform control of an unbounded, subexponential empirical process. We note here that the requirement that $\gamma_1(G) \leq 1$ in Condition 3.1(b) is precisely the condition needed to ensure uniform concentration is possible over the restricted space $\mathbb{D}_G$.

The proof of Theorem 4.1 is more subtle and involved. Since we no longer assume we can restrict to a superstructure, uniform score concentration (i.e. over the full space $\mathbb{D}$) is no longer readily viable. As a result, we must obtain *uniform control* over all $p2^p$ neighbourhood regression problems. Let $\beta_j(S) = \Sigma_{SS}^{-1}\Sigma_{Sj}$ denote the population regression coefficients of $X_j$ onto $X_S$, where $S \subset [p]_j$. It is not hard to show that $\widehat{B}$ reduces to estimating $\beta_j(S)$ for $p$ *random* sets $S$ that depend on $\mathbf{X}$ with the penalized least-squares estimator

$$\widehat{\beta}_j(S) \in \underset{\theta \in \mathbb{R}^m, \ \mathrm{supp}(\theta) \subset S}{\arg\min} \ \frac{1}{2n}\|\mathbf{x}_j - \mathbf{X}\theta\|_2^2 + \rho_\lambda(\theta).$$

It turns out that these estimators have a great deal of redundancy, and in order to control all $p2^p$ such estimators, it suffices to control at most $O(p^d)$ of them. In order to prove this, we show that the following set system has a largest element $M_j(S)$ (Lemma B.2):

$$\mathcal{T}_j(S) = \{T \subset [p]_j : \ \beta_j(T) = \beta_j(S)\}.$$

Let $M_j(S)$ be this largest element, i.e. $T \in \mathcal{T}_j(S) \implies T \subset M_j(S)$. Then there are at most $O(p^d)$ such sets, and we show that in order to control $\beta_j(S)$ for all $S$, it suffices to control each $\beta_j(M_j(S))$ (Corollary B.4). The final piece of the proof is to establish control over $\rho_\lambda(\widehat{B})$; this follows from a somewhat lengthy but straightforward Gaussian concentration argument.

# 6 Discussion

We have established that a score-based estimator achieves $\Omega(s \log p)$ sample complexity for learning a sparse, minimum-trace DAG, and extended these results to the nonidentifiable setting. The proof technique is novel, leveraging the lattice structure of Gaussian conditional independence. Compared to recent theoretical work on DAG learning that sidesteps optimization altogether, our approach directly attacks a difficult nonconvex optimization problem. To conclude this paper, we discuss some limitations, extensions, and directions for future research.

**Computation** Since (1) is a nonconvex program, computation of $\widehat{B}$ is challenging and in fact NP-hard [7]. Fortunately, there are fast algorithms via dynamic programming for finding globally optimal Bayesian networks [43, 55, 56]. For example, by combining dynamic programming with A* search, Xiang and Kim [68] propose an exact algorithm to compute the $\ell_1$-regularized version of $\widehat{B}$ that is tractable on problems with hundreds of nodes. More recently, a mixed-integer formulation has also been proposed [9, 10]. Recent work [77] has also shown that the program (1) can be solved approximately with second-order methods, and the resulting solutions are often very close to the true global minimum in practice. Given the NP-hardness of computing $\widehat{B}$, an important direction for future work is to determine whether or not there exists a polynomial-time estimator that can achieve $s \log p$ sample complexity or better. As such, the current work provides important theoretical justification for this inquiry.

**Comparison to existing methods** Despite the long history of score-based methods for learning DAGs, very little is known about the explicit, finite-sample behaviour of these methods. We have already acknowledged that the estimator (1) has appeared previously in the literature without a rigorous theoretical analysis [e.g. 26, 51, 54, 68, 77]. The well-known GES algorithm, on the other hand, has asymptotic consistency guarantees in both the low- [6] and high-dimensional [40] settings. We do not pursue a detailed experimental comparison of these two popular approaches here for the simple reason that this has already been done, see e.g. [1, 68, 70, 77]. These papers indicate that even approximate algorithms for $\widehat{B}$ outperform GES (along with other algorithms such as PC and MMHC) on a wide variety of settings and graphs.

**Comparison to nonconvex models in ML**    Much of the interest in the current work stems not only from providing explicit finite-sample guarantees for the DAG learning problem, but also from its analysis of a highly nonconvex optimization problem. For this reason, it is worth comparing our results with recent work on nonconvex models in the ML literature [5, 8, 13, 14, 16, 18, 19, 28–30, 33, 38, 58, 60]. In particular, we note the spate of recent papers on so-called "benign nonconvexity", which is the idea that although a problem may be nonconvex, its geometry is such that the nonconvexity is not a practical issue. Conditions ensuring this include the Polyak-Lojasiewicz condition [32], restricted strong convexity [36], and "strict" or "rideable" saddle points [17, 59]. Unfortunately, this approach of benign nonconvexity does not apply to optimizing (1) since this problem is easily shown to violate these properties, and in particular, there exist local minima that are not global. While this may seem discouraging, we note that recent work [77] has shown that second-order algorithms often find the global minimum in practice. We leave it to future work to study this behaviour in more detail.

### Acknowledgments

We thank the anonymous reviewers for their feedback. The authors acknowledge the support of the NSF via IIS-1546098.

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
