[Supplementary Material]


## Supplementary Material for: Globally optimal score-based learning of directed acyclic graphs in high-dimensions

Appendix A contains some background and preliminary material that is important for the proof. We then use Appendix B to outline the main ideas and postpone detailed proofs of the various technical results to Appendices C-D. The reader interested in skipping directly to the proofs of the main theorems can find them in Appendices B.5 (Theorem 3.1) and B.6 (Theorem 4.1).

## A Preliminaries

We begin by reviewing the connection between the equivalence class $\mathfrak{D}(\Sigma)$, Cholesky factors of $\Sigma$, and permutations. This material is essential and forms the basis for our proof technique. We then give some details on the regularizers used, and conclude with some important definitions on neighbourhood regression problems and introduce the concept of a *model selection exponent*.

### A.1 Permutations

Denote the class of permutations on $p$ elements by $\mathbb{S}_p$. For each $\pi \in \mathbb{S}_p$, define the associated permutation operator $P_\pi$ on matrices: For any matrix $A$, $P_\pi A$ is the matrix obtained by permuting the rows and columns of $A$ according to $\pi$, so that $(P_\pi A)_{ij} = a_{\pi(i)\pi(j)}$.

**Cholesky representation**    Fix $\pi \in \mathbb{S}_p$. Write $\Gamma := \Sigma^{-1}$ and use the (modified) Cholesky decomposition to write $P_\pi \Gamma = (I - L)D^{-1}(I - L)^T$ where $L$ is strictly lower triangular and $D \in \mathbb{R}^p_+$. Define $\widetilde{B}(\pi) := P_{\pi^{-1}}L$ and $\widetilde{\Omega}(\pi) := P_{\pi^{-1}}D$. The following result is well-known, but is restated here for completeness:

**Lemma A.1.** *For any $\Sigma \succ 0$, the equivalence class of $\Sigma$ (cf. (4)) is $\mathfrak{D}(\Sigma) = \{\widetilde{B}(\pi) : \pi \in \mathbb{S}_p\}$.*

Thus we can always write an arbitrary element of $\mathfrak{D}(\Sigma)$ as $\widetilde{B}(\pi)$. The permutation $\pi$ represents a valid topological sort for $\widetilde{B}(\pi)$. The columns of $\widetilde{B}(\pi)$ will be denoted by $\widetilde{\beta}_j(\pi)$, and the $j$th diagonal element of $\widetilde{\Omega}(\pi)$ will be denoted by $\widetilde{\omega}_j^2(\pi)$. It follows from these definitions and (2) that

$$X_j = \widetilde{\beta}_j(\pi)^T X + \widetilde{\varepsilon}_j(\pi), \quad \text{where} \quad \widetilde{\varepsilon}_j(\pi) \sim \mathcal{N}(0, \widetilde{\omega}_j^2(\pi)), \tag{15}$$

for $j = 1, \ldots, p$. Note that $\mathrm{supp}(\widetilde{\beta}_j(\pi)) \subset S_j(\pi)$ for all $j = 1, \ldots, p$, where

$$S_j(\pi) := \{k : \pi^{-1}(k) > \pi^{-1}(j)\} \tag{16}$$

consists of the nodes $X_k$ that come after $X_j$ under the ordering $X_{\pi(i)} \prec X_{\pi(i+1)}$ for $i = 1, \ldots, p-1$.

**Minimum-trace permutations**    In Section 2.1, we defined the notion of a minimum-trace DAG $\widetilde{B}_{\min}$. By Lemma A.1, we know that $\widetilde{B}_{\min} = \widetilde{B}(\pi)$ for some $\pi \in \mathbb{S}_p$, where $\pi$ is not necessarily unique. This motivates the following definition:

*Definition* A.1.  The set of *minimum-trace permutations* is defined to be

$$\Pi_0 := \underset{\pi \in \mathbb{S}_p}{\arg\min} \ \mathrm{tr}\,\widetilde{\Omega}(\pi). \tag{17}$$

Given $\pi_0 \in \Pi_0$, $\pi_0$ is called a *minimum-trace permutation* and the corresponding DAG $\widetilde{B}(\pi_0)$ is called a *minimum-trace DAG*. This definition does not require that $\widetilde{B}_{\min}$ is unique, and allows for the possibility that $\widetilde{B}(\pi_1) \neq \widetilde{B}(\pi_2)$ for $\pi_1, \pi_2 \in \Pi_0$.

**Estimated permutations**    Recall that $\mathbb{D}$ is the space of $p \times p$ real matrices that represent DAGs when interpreted as weighted adjacency matrices. For each $\pi \in \mathbb{S}_p$, define

$$\mathbb{D}[\pi] = \{B \in \mathbb{D} : P_\pi B \text{ is lower triangular}\}. \tag{18}$$

A DAG $B = [\,\beta_1\,|\,\cdots\,|\,\beta_p\,] \in \mathbb{D}$ is in $\mathbb{D}[\pi]$ if and only if $\mathrm{supp}(\beta_j) \subset S_j(\pi)$ for all $j = 1, \ldots, p$. In other words, for each node $X_j$, the permutation $\pi$ defines a unique set of candidate parents given

by (16), and $B \in \mathbb{D}[\pi]$ if and only if the parent set of $\beta_j$ comes from $S_j(\pi)$ for all $j$. By definition, $\widetilde{B}(\pi) \in \mathbb{D}[\pi]$ for every $\pi$ and hence $\text{supp}(\widetilde{\beta}_j(\pi)) \subset S_j(\pi)$ for all $j$.

Recall the estimator $\widehat{B}$ defined via (1). The following definition formalizes the collection of permutations that are topological sorts for $\widehat{B}$:

*Definition* A.2. The *collection of estimated permutations* is

$$\widehat{\Pi} = \{\pi \in \mathbb{S}_p : \widehat{B} \in \mathbb{D}[\pi]\}.$$

An arbitrary element of $\widehat{\Pi}$ will be denoted by $\widehat{\pi}$. An equivalent definition of $\widehat{\Pi}$ is the set of permutations $\pi$ such that $P_\pi \widehat{B}$ is lower triangular.

## A.2 Regularizers

We study both the $\ell_1$ and MCP regularizers as given by Condition 2.1. Here we summarize some properties of these regularizers for later use.

**Lemma A.2.** *Suppose $\rho_\lambda$ is either $\ell_1$ or MCP. Then $\rho_\lambda$ satisfies the following conditions:*

  *(a) $\rho_\lambda$ is concave and nondecreasing;*

  *(b) $\rho_\lambda(0) = 0$;*

  *(c) There are constants $\underline{\rho}_0, \underline{\rho}_1 \geq 0$, independent of $\lambda$, such that $\rho_\lambda(x) \geq \min\{\underline{\rho}_1 \lambda x, \underline{\rho}_0 \lambda^2\}$.*

**Lemma A.3.** *Suppose $\rho_\lambda$ is either $\ell_1$ or MCP. Then $\rho_\lambda$ is additionally right-differentiable at zero and satisfies $0 < \rho'_\lambda(0+) < \infty$.*

An elementary consequence of Lemma A.1 is that $\rho_\lambda$ is subadditive. Lemma A.1(c) says that $\rho_\lambda$ can be bounded below by a capped-$\ell_1$ penalty: It is always true that a concave, nondecreasing function can be bounded below by a capped-$\ell_1$ penalty, and Lemma A.1(c) simply normalizes this capped-$\ell_1$ penalty in terms of $\lambda$.

For completeness, we summarize below both regularizers under consideration along with the constants involved in the previous lemmas.

– The minimax concave penalty (MCP) proposed by Zhang [72]:

$$\rho_\lambda(x; \gamma) := \lambda\Big(x - \frac{x^2}{2\lambda\gamma}\Big)1(x < \lambda\gamma) + \frac{\lambda^2\gamma}{2}1(x \geq \lambda\gamma). \tag{19}$$

The MCP has $\rho'_\lambda(0+) = \lambda$, $\underline{\rho}_1 = 1/2$, and $\underline{\rho}_0 = \gamma/2$.

– The $\ell_1$ penalty, $\rho_\lambda(x) = \lambda x$, has $\rho'_\lambda(0+) = \lambda$, $\underline{\rho}_1 = 1$, and $\underline{\rho}_0 \in [0, \infty)$.

Finally, since several of the results proved in this supplement do not require the incoherence condition $\zeta(G) < 1$, we will also make use of the following weaker version of Condition 2.1:

**Condition A.1** (Regularizer)**.** *The regularizer $\rho_\lambda$ is chosen to be $\ell_1$ or the MCP.*

## A.3 Neighbourhood regression

The core of our analysis is the regression decomposition (15) which we interpret as a neighbourhood regression problem and is used to learn the parent set of a node and hence the DAG structure. In this section we formalize these notions and introduce the concept of a *model selection exponent*, which quantifies the difficulty of a neighbourhood regression problem.

### A.3.1 Penalized least-squares estimators

We are interested in the population SEM coefficients given by the following:

*Definition* A.3. For any $S \subset [p]_j$, let

$$\beta_j(S) := \underset{\beta \in \mathbb{R}^p, \; \text{supp}(\beta) \subset S}{\arg\min} \mathbb{E}\big[X_j - \beta^T X\big]^2.$$

We call $\beta_j(S)$ the SEM coefficients for $X_j$ and denote the support set of $\beta_j(S)$ by $m_j(S) :=$ supp$(\beta_j(S))$.

Note that $\beta_j(S) = \Sigma_{SS}^{-1}\Sigma_{Sj}$. Every positive definite matrix $\Sigma$ defines a collection of $p2^{p-1}$ SEM coefficients given by $\{\beta_j(S) : S \subset [p]_j, j \in [p]\}$. We will be interested in estimating $\beta_j(S)$ via penalized least-squares (PLS):

*Definition* A.4. Suppose $y \in \mathbb{R}^n$ and $Z \in \mathbb{R}^{n \times m}$. Let $S \subset [m]$ and consider the set defined by

$$\widehat{\Theta}_\lambda(y, Z; S) := \underset{\theta \in \mathbb{R}^m,\, \text{supp}(\theta) \subset S}{\arg\min}\; \frac{1}{2n}\|y - Z\theta\|_2^2 + \rho_\lambda(\theta), \qquad (20)$$

i.e., the set of global minimizers of the support-restricted PLS problem above. Let $\widehat{\Theta}_\lambda(y, Z) := \widehat{\Theta}_\lambda(y, Z; [m])$ correspond to the case where there is no support restriction.

The support-restricted PLS problem $\widehat{\Theta}_\lambda(y, Z; S)$ allows us to properly define a neighbourhood regression problem. Let $\mathbf{x}_j$ denote the $j$th column of $\mathbf{X}$.

*Definition* A.5 (Neighbourhood regression). The *neighbourhood regression problem* for node $X_j$ given a neighbourhood $S \subset [p]_j$ is defined to be the (possibly nonconvex) program given by $\widehat{\Theta}_\lambda(\mathbf{x}_j, \mathbf{X}; S)$. An arbitrary solution to this program will be denoted by $\widehat{\beta}_j(S)$, i.e. $\widehat{\beta}_j(S) \in \widehat{\Theta}_\lambda(\mathbf{x}_j, \mathbf{X}; S)$.

Learning $\widehat{B}$ reduces to controlling $\widehat{\Theta}_\lambda(\mathbf{x}_j, \mathbf{X}; S)$ for specific choices of $S$ for each $j$ (Lemma B.1). Thus, in the sequel, we no longer need to consider individual permutations, and instead will restrict our attention to subsets $S$, called candidate sets.

### A.3.2 Model selection exponents

Given some $n \times m$ matrix $Z$ and $m$-vector $\theta^*$, define a set of "bad" noise vectors as follows:

$$A(Z, \theta^*; S) := \big\{\, w \in \mathbb{R}^n : \text{supp}(\widehat{\theta}) \neq \text{supp}(\theta^*) \qquad (21)$$
$$\text{for some } \widehat{\theta} \in \widehat{\Theta}_\lambda(Z\theta^* + w, Z; S) \,\big\}.$$

For a random vector $\mathbf{w} \in \mathbb{R}^n$ (e.g. $\mathbf{w} \sim \mathcal{N}_n(0, \sigma^2 I_n)$), we then have the following model selection failure event:

$$\mathcal{A}(\mathbf{w}, Z, \theta^*; S) := \big\{\mathbf{w} \in A(Z, \theta^*; S)\big\}. \qquad (22)$$

As usual we use the shorthand $\mathcal{A}(\mathbf{w}, Z, \theta^*) = \mathcal{A}(\mathbf{w}, Z, \theta^*; [m])$.

*Definition* A.6. Given a regularizer $\rho_\lambda$, the *model selection exponent* for the regression problem $\mathbf{y} = Z\theta^* + \mathbf{w}$ is defined to be

$$\Phi_\lambda(Z, \theta^*, \sigma^2) := -\log \mathbb{P}\big[\mathcal{A}(\mathbf{w}, Z, \theta^*)\big],$$

where $\mathbb{P}$ is taken with respect to the distribution of $\mathbf{w} \sim \mathcal{N}_n(0, \sigma^2 I_n)$.

A larger exponent corresponds to better model selection performance. Let $\sigma_{\max}^2 := \max_{1 \leq j \leq p} \text{var}(X_j)$ and note that $\sigma_{\max}^2 \leq r_{\max}(\Sigma)$. Define

$$\Psi_\lambda = \Psi_\lambda(\mathbf{X}) := \inf_{\substack{0 < \sigma \leq \sigma_{\max}}} \inf_{\substack{\|\theta\|_0\, \leq\, d(\Sigma) \\ \tau_*(\theta)\, \geq\, \tau_*(\Sigma)}} \Phi_\lambda(\mathbf{X}, \theta, \sigma^2). \qquad (23)$$

This quantity measures "how difficult" the model selection problems defined by the fixed matrix $\mathbf{X}$ are, and encodes what is usually proved in the regression literature: An upper bound on the probability of model selection failure given the maximum sparsity level $d$, minimum signal strength $\tau_*$, and the maximum variance $\sigma_{\max}^2$. This probability generally depends on the regularization parameter $\lambda$, which in turn may depend on any of these quantities.

 **A.3.3 Example model selection exponents**

607 To illustrate, let us derive a model selection exponent for the MCP, defined in (19). Huang et al. [27]
608 consider PLS estimators $\widehat{\Theta}_\lambda(y, Z; S)$ as defined in (20), applied to the data from a linear regression
609 model $y = Z\theta^* + \mathbf{w}$, and provides conditions for model selection consistency. Adapting their result
610 to our setup and notation, we have the following bound on model selection exponent for the MCP:

611 **Lemma A.4.** *Suppose* $\mathbf{X} \overset{iid}{\sim} \mathcal{N}_p(0, \Sigma)$*. Take* $\rho_\lambda = \rho_\lambda(\,\cdot\,; \gamma)$ *as in* (19) *and assume* $\Sigma$ *is positive*
612 *definite with bounded eigenvalues. Assume that*

613      *1.* $d(\Sigma) \leq \kappa_4 \cdot \min\{p, n, n/\log p\}$,

614      *2.* $\tau_*(\Sigma) > (1 + \gamma)\lambda$ *for some* $\gamma > \kappa_5 > 0$.

615 *Then for any* $\lambda \geq \kappa_6 \cdot \sqrt{(d+1)\log p/n}$*, it follows that* $\mathbb{E}e^{-\Psi_\lambda(\mathbf{X}, \Sigma)} \leq 3\exp\left(-2\min\{d\log p, n\}\right)$.
616 *Here,* $\kappa_j = \kappa_j(\Sigma)$ $(j = 4, 5, 6)$ *are constants depending only on* $\{r_{\min}(\Sigma), r_{\max}(\Sigma)\}$.

617 This lemma is a straightforward consequence of Theorem 4.2 in [27] and Proposition 2 in [73].
618 Briefly, [27] show that the least-squares MCP estimator correctly recovers the support of a linear
619 model as long as the so-called *sparse Riesz condition* holds. We then use [73] to bound the probability
620 that $\mathbf{X}$ satisfies this condition. For the special case $\beta_j(S) = 0$ (which is not covered by [27]) we can
621 invoke Proposition D.4.

622 In a similar manner, analogous bounds can be derived for other regularizers using existing results, see
623 e.g. [27, 36, 66]. For example, using Corollary 1(a) in [36], a similar bound for $\ell_1$-regularization can
624 be derived under the additional assumption that $\zeta(G) < 1$ as long as $n \gtrsim d\log p$.

# B    Outline of proofs

626 We seek control over the following event:

$$\mathcal{B} := \{\operatorname{supp}(\widehat{B}) \neq \operatorname{supp}(\widetilde{B}(\widehat{\pi}))\ \exists\, \widehat{\pi} \in \widehat{\Pi}\}. \tag{24}$$

627 We will do this by reducing the analysis of $\widehat{B}$ to a family of neighbourhood regression problems.
628 There are two key steps: (i) Showing that $\widehat{B}$ is equivalent to solving a series of $p$ *random* regression
629 problems given by $\widehat{\Theta}_\lambda(\mathbf{x}_j, \mathbf{X}; S_j(\widehat{\pi}))$ (cf. Definition A.5), and (ii) Controlling the neighbourhood
630 problems $\widehat{\Theta}_\lambda(\mathbf{x}_j, \mathbf{X}; S_j(\widehat{\pi}))$ for all $\widehat{\pi} \in \widehat{\Pi}$.

631 The second step (ii) highlights the main technical difference between Theorems 3.1 and 4.1:

632     • To prove Theorem 3.1, we first prove a uniform concentration result for the score $Q(B)$, and
633        use this to show that $\widehat{\Pi} \subset \Pi_0$. That is, any estimated permutation must be a minimum-trace
634        permutation. As a result, the random permutations $\widehat{\pi} \in \widehat{\Pi}$ are confined to live in a small set,
635        which makes controlling the neighbourhood problems simpler. As a result we are able to
636        bound $\mathbb{P}(\operatorname{supp}(\widehat{B}) \neq \operatorname{supp}(\widetilde{B}(\widehat{\pi})))$ directly, which implies bounds on the desired quantities
637        with $\widehat{\pi}$ replaced by a minimum-trace permutation $\pi_0$.

638     • To prove Theorem 4.1, we no longer assume we can restrict to a superstructure, and hence
639        uniform score concentration (i.e. over the full space $\mathbb{D}$) is no longer readily viable. As a
640        result, we must obtain *uniform control* over the neighbourhood problems $\widehat{\Theta}_\lambda(\mathbf{x}_j, \mathbf{X}; S)$ for
641        all $S$ and $j$. The challenge is that there are superexponentially many regression problems, so
642        a naïve union bound over this family would yield overly pessimistic bounds on the order
643        $p/n$. To deal with this, we will exploit a lattice property of these problems.

644 The proofs of Theorem 3.1 and 4.1 will be broken down into several steps. First, we establish
645 some basic properties of the objective function and the probability space in order to reduce the
646 neighbourhood regression analysis to a family of maximal sets denoted by $M_j(S)$ (Definition B.2).
647 Then we introduce the lattice property (Lemmas B.2 and B.3) that is central to our proofs, and exploit
648 this to provide a uniform bound on the probability of false selection for any neighbourhood problem
649 (Proposition B.5).

## B.1 Reduction to neighbourhood regression

Recall that the $j$th column of $\widehat{B}$ is denoted by $\widehat{\beta}_j$ and denote the sample version of $\widetilde{\varepsilon}_j(\pi)$ by boldface, i.e. $\widetilde{\boldsymbol{\varepsilon}}_j(\pi) := \mathbf{x}_j - \mathbf{X}\widetilde{\beta}_j(\pi)$. The first step above is justified by the following result. The symbol $\perp\!\!\!\perp$ is used here to denote independence of random variables.

**Lemma B.1.** *Suppose* $\mathbf{X} \overset{iid}{\sim} \mathcal{N}_p(0, \Sigma)$ *and* $\lambda \geq 0$. *Then the following statements are true:*

(a) *For any* $j \in [p]$ *and* $\pi \in \mathbb{S}_p$, $\widetilde{\boldsymbol{\varepsilon}}_j(\pi) \perp\!\!\!\perp \mathbf{X}_{S_j(\pi)}$.

(b) $\widehat{B}$ *is a global minimizer of* $Q(B)$ *if and only if* $\widehat{\beta}_j \in \widehat{\Theta}_\lambda(\mathbf{x}_j, \mathbf{X}; S_j(\widehat{\pi}))$ *for each* $j \in [p]$ *and* $\widehat{\pi} \in \widehat{\Pi}$.

The proof of this lemma, which is a simple consequence of how the least-squares loss and the regularizer factor, is found in Appendix C.3. This allows us to formally establish the equivalence between the DAG problem and neighbourhood regression: In order to construct $\widehat{B}$, it suffices to solve a neighbourhood regression problem for each column of $\widehat{B}$, given by $\widehat{\Theta}_\lambda(\mathbf{x}_j, \mathbf{X}; S_j(\widehat{\pi}))$. A key observation is that through the independence established in Lemma B.1(a) and a conditioning argument, we can reduce the regression problem given by $\widehat{\Theta}_\lambda(\mathbf{x}_j, \mathbf{X}; S_j(\widehat{\pi}))$ to a fixed design problem. The details are outlined in the proof of Proposition B.5.

## B.2 Invariant sets and monotonicity

As a consequence of Lemma B.1, we have (cf. (24))

$$\mathcal{B} \subset \{\text{supp}(\widehat{\beta}_j(S)) \neq \text{supp}(\beta_j(S)) \ \exists j \in [p], S \subset [p]_j\}. \tag{25}$$

In order to further reduce the total number of estimators we must control, we will introduce the notion of an *invariant set*. First, recall the definition of $\beta_j(S)$ (cf. Definition A.3) and for any $j \in [p]$ and $S \subset [d]_j$ define the error (or noise) for the associated neighbourhood regression as the following residual:

$$\varepsilon_j(S) := X_j - \beta_j(S)^T X.$$

The support set of $\beta_j(S)$ is denoted by $m_j(S) := \text{supp}(\beta_j(S))$ and the error variance by $\omega_j^2(S) := \text{var}(\varepsilon_j(S))$.

*Definition* B.1. For any $S \subset [p]_j$, define a collection of subsets by

$$\mathcal{T}_j(S) := \{T \subset [p]_j : \beta_j(T) = \beta_j(S)\} = \{T \subset [p]_j : m_j(T) = m_j(S)\},$$

where $\beta_j(S)$ and $m_j(S)$ are defined in Definition A.3. If $T \in \mathcal{T}_j(S)$, we call $T$ an *invariant set of $S$ for $j$*, or *$S$-invariant* for short.

In other words, for any $j$, $\mathcal{T}_j(S)$ is the collection of candidate sets $T \subset [p]_j$ such that the projection of $X_j$ onto $\{X_i, i \in T\}$ is invariant. With some abuse of terminology, let us refer to $m_j(T) = \text{supp}(\beta_j(T))$ as the *support of neighbourhood $T$* (for node $j$). An equivalent description of $\mathcal{T}_j(S)$ is the set of neighbourhoods $T$ whose support (for node $j$) is the same and equals $m_j(S)$.

The following lemma illustrates a crucial property of invariant sets:

**Lemma B.2.** $T_1, T_2 \in \mathcal{T}_j(S) \implies T_1 \cup T_2 \in \mathcal{T}_j(S)$.

This justifies the following definition:

*Definition* B.2. The unique largest element of $\mathcal{T}_j(S)$ shall be denoted by $M_j(S)$. Formally,

$$M_j(S) := \bigcup \mathcal{T}_j(S) = \bigcup \{T \subset [p]_j : \beta_j(T) = \beta_j(S)\}.$$

The name "$S$-invariant set" comes from the fact that for any $T \in \mathcal{T}_j(S)$, we have the following useful identities:

$$\beta_j(m_j(S)) = \beta_j(S) = \beta_j(T) = \beta_j(M_j(S)), \tag{26}$$
$$\varepsilon_j(m_j(S)) = \varepsilon_j(S) = \varepsilon_j(T) = \varepsilon_j(M_j(S)). \tag{27}$$

The reason for introducing invariant sets is that it is generally *sufficient* to study the neighbourhood problem for $M_j(S)$ in the sense that once we have model selection consistency for each estimator in $\widehat{\Theta}_\lambda(\mathbf{x}_j, \mathbf{X}; M_j(S))$, the same is *guaranteed* for estimators based on every other neighbourhood in $\mathcal{T}_j(S)$. In fact, we have the following result, which says that model selection properties of the $S$-restricted estimators are monotone with respect to those sets $S$ that contain the true support.

**Lemma B.3.** *Suppose that $Z \in \mathbb{R}^{n \times m}$ is fixed and consider the regression problem $\mathbf{y} = Z\theta^* + \mathbf{w}$ for some $\theta^* \in \mathbb{R}^m$. If $\operatorname{supp}(\theta^*) \subset S \subset U$, then we have the following inclusion: $A(Z, \theta^*; S) \subset A(Z, \theta^*; U)$. In particular, $\mathcal{A}(\mathbf{w}, Z, \theta^*; S) \subset \mathcal{A}(\mathbf{w}, Z, \theta^*; U)$ where $A(Z, \theta^*; S)$ and $\mathcal{A}(\mathbf{w}, Z, \theta^*; S)$ are defined in (21)–(22).*

We are interested in the model selection failure of $\widehat{\beta}_j(S)$ for $\beta_j(S)$, which can be stated as

$$\Big\{ \operatorname{supp}(\widehat{\beta}_j(S)) \neq \operatorname{supp}(\beta_j(S))$$
$$\text{for some } \widehat{\beta}_j(S) \in \widehat{\Theta}_\lambda(\mathbf{x}_j, \mathbf{X}; S) \Big\} = \mathcal{A}(\widetilde{\varepsilon}_j(S), \mathbf{X}, \beta_j(S); S) \tag{28}$$

in the notation introduced in (22).

**Corollary B.4.** *Suppose $\mathbf{X} \overset{iid}{\sim} \mathcal{N}_p(0, \Sigma)$. For any $S \subset [p]_j$, we have*

$$\mathcal{A}\Big(\widetilde{\varepsilon}_j(S), \mathbf{X}, \beta_j(S); S\Big) \subset \mathcal{A}\Big(\widetilde{\varepsilon}_j(M_j(S)), \mathbf{X}, \beta_j(M_j(S)); M_j(S)\Big).$$

Lemma B.4 is a *deterministic* statement about the events defined in (28), and proves that in order to control the neighbourhood regression problem for some set $S \subset [p]_j$, it suffices to control the strictly harder problem given by $M_j(S)$.

## B.3 A bound on false selection

For any $\Sigma \succ 0$ and fixed node $X_j$, define the following collections of subsets:

$$m_j(\Sigma) := \{m_j(S) : S \subset [p]_j\}, \tag{29}$$
$$M_j(\Sigma) := \{M_j(S) : S \subset [p]_j\}. \tag{30}$$

Note that $|m_j(\Sigma)| = |M_j(\Sigma)|$. As long as it is clear whether the argument is a set $S$ or a matrix $\Sigma$, this should not cause any confusion with $m_j(S)$ and $M_j(S)$.

For any neighbourhood $S \subset [p]_j$, recall that the associated error variance is given by $\omega_j^2(S) = \operatorname{var}(\varepsilon_j(S))$. With some more abuse of notation, let

$$\Phi_j(S) := \Phi_\lambda(\mathbf{X}_S, (\beta_j(S))_S, \omega_j^2(S)). \tag{31}$$

Note that we must restrict the SEM coefficients $\beta_j(S)$ to the subset $S$ in order for this exponent to be well-defined. Since $\operatorname{supp}(\beta_j(S)) \subset S$, this does not change anything. The following general result gives a uniform upper bound on the probability of false selection for any neighbourhood problem in terms of the maximal sets $M_j(T)$.

**Proposition B.5.** *Fix $j \in [p]$ and $\Sigma \succ 0$. Then we have*

$$\mathbb{P}\big(\operatorname{supp}(\widehat{\beta}_j(S)) \neq \operatorname{supp}(\beta_j(S)), \, \exists\, S \subset [p]_j\big) \leq \sum_{T \in m_j(\Sigma)} \mathbb{E}e^{-\Phi_j(M_j(T))},$$

*where $m_j(\Sigma)$ is defined by (29) and $\Phi_j(\,\cdot\,)$ is defined by (31).*

The proof of this result can be found in Appendix C.7. The following result—which is proved in the course of proving Proposition B.5—will also be useful when proving Theorem 3.1:

**Corollary B.6.** *Fix $j \in [p]$, $S \subset [p]_j$ and $\Sigma \succ 0$. Then we have*

$$\mathbb{P}\big(\operatorname{supp}(\widehat{\beta}_j(T)) \neq \operatorname{supp}(\beta_j(T)), \, \exists\, T \in \mathcal{T}_j(S)\big) \leq \mathbb{E}e^{-\Phi_j(M_j(S))},$$

*where $\Phi_j(\,\cdot\,)$ is defined by (31).*

Proposition B.5 says that to control the probability of false selection uniformly for all $2^{p-1}$ neighbourhoods $S$ of the node $j$, it suffices to control a much smaller class of problems given by the neighbhourhoods $M_j(T)$ for each support set $T \in m_j(\Sigma)$.

 **B.4 Uniform support recovery**

 The following result is a key ingredient in the proofs of both Theorem 3.1 and 4.1. It establishes an
 upper bound on the probability of false selection, uniform over all $S$ and $j$.

 **Theorem B.1.** *Suppose* $\mathbf{X} \overset{iid}{\sim} \mathcal{N}_p(0, \Sigma)$ *with* $\Sigma \succ 0$*. Then*

$$\mathbb{P}\big( \operatorname{supp}(\widehat{\beta}_j(S)) = \operatorname{supp}(\beta_j(S)), \ \forall\, j \in [p],\ S \subset [p]_j \big) \geq 1 - p\binom{p}{d}\mathbb{E}e^{-\Psi_\lambda(\mathbf{X},\Sigma)}.$$

 *Proof.* For any $T \in m_j(\Sigma)$, Lemma B.3 applied with $S = M_j(T)$ and $U = [p]$ yields

$$\Phi_j(M_j(T)) \geq \Phi_\lambda(\mathbf{X},\, \beta_j(T),\, \omega_j^2(T)).$$

 Recalling $d(\Sigma)$ and $\tau_*(\Sigma)$ in Definition 4.1, we have $\|\beta_j(T)\|_0 \leq d(\Sigma)$ and $\tau_*(\beta_j(T)) \geq \tau_*(\Sigma)$, as
 well as $\omega_j^2(T) \leq \sigma_{\max}^2$. The previous expression combined with (23) implies:

$$\Phi_j(M_j(T)) \geq \Psi_\lambda(\mathbf{X},\Sigma) \quad \text{for all } T \in M_j(\Sigma). \tag{32}$$

 Combining Proposition B.5, (32) and a union bound over $j \in [p]$,

$$\mathbb{P}\big( \operatorname{supp}(\widehat{\beta}_j(S)) \neq \operatorname{supp}(\beta_j(S)), \ \exists\, j \in [p],\ S \subset [p]_j \big)$$

$$\leq \ \sum_{j=1}^{p} \sum_{T \in m_j(\Sigma)} \mathbb{E}\exp(-\Phi_j(M_j(T)))$$

$$\leq \ p\binom{p}{d} \mathbb{E}\exp(-\Psi_\lambda(\mathbf{X},\Sigma)), \tag{33}$$

 since there are at most $\binom{p}{d}$ subsets in $m_j(\Sigma)$. $\qquad\square$

 **B.5 Proof of Theorem 3.1**

 Define $\ell(B) = \|\mathbf{X} - \mathbf{X}B\|_F^2/(2n)$. There are two terms that we need to control: (i) The fluctuations
 $|\ell(B) - \mathbb{E}\ell(B)|$ and (ii) The population loss $\mathbb{E}\ell(\widehat{B})$. The fluctuations (i) are controlled by the
 following proposition, which is proved in Appendix C.8, and may be of independent interest due to
 its uniform control of an unbounded, subexponential empirical process:

 **Proposition B.7.** *Let* $\ell(B) = \|\mathbf{X} - \mathbf{X}B\|_F^2/(2n)$ *and let* $\gamma_1(G)$ *and* $\gamma_2(G)$ *be defined by* (8) *and*
 (9)*. Assume* $\gamma_1 \leq 1$*. Then there is a constant* $\kappa(\Sigma; s)$*, depending only on* $\Sigma$ *and* $s$*, such that*

$$|\ell(B) - \mathbb{E}\ell(B)| \leq \gamma_1\big[1 + 6\kappa(\Sigma; s)\gamma_2\big]\mathbb{E}\ell(B) \quad \text{for all } B \in \mathbb{D}_G \tag{34}$$

 *with probability at least* $1 - \alpha$*, where*

$$\alpha := 2p^{-1}\Big(\frac{9\gamma_1 p}{s}\Big)^{-s} + \Big(\frac{ep}{2s}\Big)^{-s}. \tag{35}$$

 For (ii), we have the following lemma:

 **Lemma B.8.** *For any* $\pi \in \mathbb{S}_p$*, we have*

$$\mathbb{E}\ell(B) \geq \mathbb{E}\ell(\widetilde{B}(\pi)) = \operatorname{tr} \widetilde{\Omega}(\pi) \quad \text{for all } B \in \mathbb{D}_p[\pi], \tag{36}$$

 *where equality holds if and only if* $B = \widetilde{B}(\pi)$*.*

 Lemma B.8 implies, in particular, that $\mathbb{E}\ell(\widehat{B}) \geq \mathbb{E}\ell(\widetilde{B}(\widehat{\pi}))$.

 By Condition 3.1(a), we have $\widetilde{B}(\pi_0) = \widetilde{B}_{\min}$ and $\widetilde{\Omega}(\pi_0) = \widetilde{\Omega}_{\min}$ for all $\pi_0 \in \Pi_0$. For any two
 permutations $\pi', \pi \in \mathbb{S}_p$ and $\eta > 0$, define a function

$$h(\pi', \pi;\, \eta) = (1 - \eta)\operatorname{tr}\widetilde{\Omega}(\pi') - (1 + \eta)\operatorname{tr}\widetilde{\Omega}(\pi) - \rho_\lambda(\widetilde{B}(\pi)). \tag{37}$$

so that we have, recalling (7),

$$\chi(\eta) := \inf_{\pi' \notin \Pi_0} \sup_{\pi \in \Pi_0} h(\pi', \pi; \eta).$$

Recall that $s$ is the maximum degree of $G$ and define $\alpha = 2p^{-1}(9\gamma_1 p/s)^{-s} + (2p/s)^{-s}$. Since $\gamma_1 \leq 1$ (and hence $n \gtrsim s \log p$), Proposition B.7 and Lemma B.8 together imply that for any $\pi \in \Pi_0$ and $\widehat{\pi} \in \widehat{\Pi}$, we have with probability $1 - \alpha$

$$
\begin{aligned}
(1 - \gamma_3)\mathbb{E}\ell(\widetilde{B}(\widehat{\pi})) + \rho_\lambda(\widehat{B}) &\leq \ell(\widehat{B}) + \rho_\lambda(\widehat{B}) \\
&\leq \ell(\widetilde{B}(\pi)) + \rho_\lambda(\widetilde{B}(\pi)) \\
&\leq (1 + \gamma_3)\mathbb{E}\ell(\widetilde{B}(\pi)) + \rho_\lambda(\widetilde{B}(\pi))
\end{aligned}
$$

where $\gamma_3 := \gamma_1[1 + 6\kappa(\Sigma; s)\gamma_2]$. Observing that $\mathbb{E}\ell(\widetilde{B}(\pi)) = \operatorname{tr}\widetilde{\Omega}(\pi)$, we thus have $h(\widehat{\pi}, \pi; \gamma_3) \leq 0$, where $h$ is given by (37).

We now show that $\widehat{\Pi} \subset \Pi_0$. Indeed, suppose $\widehat{\pi} \notin \Pi_0$ for some $\widehat{\pi} \in \widehat{\Pi}$. Then by Condition 3.1(b), $\chi(\gamma_3) = \chi(\eta) > 0$, whence

$$\sup_{\pi \in \Pi_0} h(\widehat{\pi}, \pi; \gamma_3) \geq \inf_{\pi' \notin \Pi_0} \sup_{\pi \in \Pi_0} h(\pi', \pi; \gamma_3) = \chi(\gamma_3) > 0,$$

which contradicts $h(\widehat{\pi}, \pi; \gamma_3) \leq 0$. Thus $\widehat{\Pi} \subset \Pi_0$.

By Lemma B.1(b) it suffices to show that

$$\mathbb{P}\left( \operatorname{supp}(\widehat{\beta}_j(S_j(\widehat{\pi}))) \neq \operatorname{supp}(\beta_j(S_j(\widehat{\pi}))) \, \exists\, j \in [p] \right) = O(e^{-k \log p}). \tag{38}$$

Since the minimum-trace DAG is unique, for any $\pi, \pi' \in \Pi_0$, it follows that $m_j(S_j(\pi)) = \operatorname{supp}(\widetilde{\beta}_{\min,j}) = m_j(S_j(\pi'))$ and hence $M_j(S_j(\pi)) = M_j(S_j(\pi'))$. Using $\widehat{\Pi} \subset \Pi_0$, we have

$$
\begin{aligned}
\mathbb{P}\big( \operatorname{supp}(\widehat{\beta}_j(&S_j(\widehat{\pi}))) \neq \operatorname{supp}(\beta_j(S_j(\widehat{\pi}))) \, \exists\, j \in [p] \big) \\
&\leq \mathbb{P}\left( \operatorname{supp}(\widehat{\beta}_j(S_j(\pi_0))) \neq \operatorname{supp}(\beta_j(S_j(\pi_0))) \, \exists\, j \in [p], \exists\, \pi_0 \in \Pi_0 \right) \\
&= \mathbb{P}\left( \operatorname{supp}(\widehat{\beta}_j(S_j(\pi_0))) \neq \operatorname{supp}(\widetilde{\beta}_{\min,j}) \, \exists\, j \in [p], \exists\, \pi_0 \in \Pi_0 \right) \\
&\leq \sum_{j=1}^{p} \mathbb{E}e^{-\Phi_j(M_j(\operatorname{supp}(\widetilde{\beta}_{\min,j})))},
\end{aligned}
$$

where we used Corollary B.6 in the last line. Finally, apply known bounds (see Appendix A.3.3) to deduce $\Phi_j(M_j(\operatorname{supp}(\widetilde{\beta}_{\min,j}))) \gtrsim k \log p$ whenever $n \gtrsim k \log p$, which is implied since $k \leq s$ and we have assumed already that $n \gtrsim s \log p$. (If $\ell_1$-regularization is used, this is where we also need to assume $\zeta(G) < 1$ in Condition 2.1.) This implies the desired results with probability

$$1 - 2p^{-1}\left(\frac{9\gamma_1 p}{s}\right)^{-s} - \left(\frac{2p}{s}\right)^{-s} - O(e^{-k \log p}) = 1 - O(e^{-k \log p}),$$

where we used $k \leq s$ to simplify the probability bound. This completes the proof.

## B.6  Proof of Theorem 4.1

The support recovery claim follows immediately from (25), Theorem B.1, and known bounds on the support recovery properties of penalized regression (see Section A.3.3 for discussion). Thus it remains to control $\rho_\lambda(\widehat{B})$ and $\rho_\lambda(\widetilde{B}(\widehat{\pi}))$ by $\rho_\lambda(\widetilde{B}(\pi_0))$.

The first step is the following lemma, which is a version of the standard basic inequality adapted to the current setting:

**Lemma B.9.** *Let* $\mathbf{E}(\pi) := \mathbf{X} - \mathbf{X}\widetilde{B}(\pi)$*. For any* $\pi \in \mathbb{S}_p$ *and* $\widehat{\pi} \in \widehat{\Pi}$*,*

$$
\begin{aligned}
\frac{1}{2n}\|\mathbf{X}(\widetilde{B}(\widehat{\pi}) - \widehat{B})\|_F^2 + \rho_\lambda(\widehat{B}) \;\leq\; & \frac{1}{2n}\|\mathbf{E}(\pi)\|_F^2 - \frac{1}{2n}\|\mathbf{E}(\widehat{\pi})\|_F^2 \\
& + \frac{1}{n}\operatorname{tr}\left(\mathbf{E}(\widehat{\pi})^T\mathbf{X}(\widetilde{B}(\widehat{\pi}) - \widehat{B})\right) \\
& + \rho_\lambda(\widetilde{B}(\pi)).
\end{aligned} \tag{39}
$$

The proof of Lemma B.9 can be found in Appendix C.10. Lemma B.9 helps to reduce the analysis to three terms:

(B.9a) The difference in residuals $\|\mathbf{E}(\pi)\|_F^2/(2n) - \|\mathbf{E}(\widehat{\pi})\|_F^2/(2n)$ explains the origin of the minimum-trace permutation: We would like to make $\|\mathbf{E}(\pi)\|_F^2/(2n)$ as small as possible in order to minimize this difference. By standard concentration arguments, $\|\mathbf{E}(\pi)\|_F^2/n$ is close to its expectation, $\operatorname{tr}\widetilde{\Omega}(\pi)$. Hence, we choose $\pi$ to minimize $\operatorname{tr}\widetilde{\Omega}(\pi)$. The details of this argument are in Appendix D.3; the explicit upper bound we use is detailed in Proposition D.8.

(B.9b) The quantity $\operatorname{tr}(\mathbf{E}(\widehat{\pi})^T\mathbf{X}(\widetilde{B}(\widehat{\pi}) - \widehat{B}))/n$ can be bounded using the Gaussian width condition (Definition D.1). There is a subtlety regarding whether to decompose this along rows or columns; see Lemma D.6.

(B.9c) The penalty on $\widehat{B}$ can be replaced with $\rho_\lambda(\widetilde{B}(\widehat{\pi}))$ by showing that $\rho_\lambda(\widehat{B}) \gtrsim \rho_\lambda(\widetilde{B}(\widehat{\pi}))$ (Lemma D.7).

Once we have establish control of these three terms (the details of which are found in Appendix D), we can prove the following bound in terms of the constants $\delta$ (cf. Definition D.1) and $a_2$ (cf. Condition 4.1):

**Proposition B.10.** *Assume* $n > 8\,(d+1)\log p$*. Under Condition A.1 on* $\rho_\lambda$*, further assume*

$$
\tau_*(\mathfrak{D}(\Sigma)) \geq \tau_\lambda\left(\frac{2(1+\delta)}{1-3\delta}\right) \quad \text{for some } \delta \in (0, 1/3).
$$

*Let* $\widetilde{B}_{\min} = \widetilde{B}(\pi_0)$ *be a minimum-trace DAG satisfying Condition 4.1. Then*

$$
\frac{2\delta}{1-\delta}\rho_\lambda(\widetilde{B}(\widehat{\pi})) \;\overset{(i)}{\leq}\; \rho_\lambda(\widehat{B}) \;\overset{(ii)}{\leq}\; \frac{2}{1-\delta}\left(1 + \frac{10}{a_2}\right)\rho_\lambda(\widetilde{B}(\pi_0)), \tag{40}
$$

*with probability at least* $1 - c_1 e^{-c_2 \min\{n,\,(d+1)\log p\}} - p\binom{p}{d}\mathbb{E}e^{-\psi_\lambda(\mathbf{X}, \sigma_{\max}^2; \delta)}$*.*

The proof of Proposition B.10 follows from a series of standard concentration arguments (Appendix D), and can be found in Appendix D.4.

Finally, the desired bounds on $\rho_\lambda(\widehat{B})$ and $\rho_\lambda(\widetilde{B}(\widehat{\pi}))$ follow from Proposition B.10 by taking $\delta = (a_1 - 2)/(3a_1 + 2) \in (0, 1/3)$, and using Proposition D.3 to complete the probability bound.

# C  Proofs of technical results

## C.1  Proof of Lemma 2.1

Consider the following program:

$$
\min \sum_{j=1}^{p} x_j^2 \text{ subject to } \sum_{j=1}^{p}\log x_j^2 = C. \tag{41}
$$

The solution to this program is given by $x_j^2 = e^{C/p}$ for all $j = 1, \ldots, p$. In other words, the minimum is attained by a constant vector. It is straightforward to verify that $\log\det\widetilde{\Omega}(\pi) = \log\det\Sigma$ and hence $\log\det\widetilde{\Omega}(\pi) = \sum_j \log\widetilde{\omega}_j^2(\pi)$ is constant for all $\pi \in \mathbb{S}_p$. Thus for any $\pi \in \mathbb{S}_p$, the vector $(\widetilde{\omega}_1^2(\pi), \ldots, \widetilde{\omega}_p^2(\pi)) \in \mathbb{R}^p$ is feasible for (41), which implies that $\operatorname{tr}\widetilde{\Omega}(\pi)$ is minimized whenever $\widetilde{\omega}_1^2(\pi) = \cdots = \widetilde{\omega}_p^2(\pi)$. Finally, uniqueness of $\widetilde{B}(\pi_0)$ follows from Theorem 1 in Peters and Bühlmann [47].

## C.2 Proof of Lemma A.1

We need the following simple lemma, which follows since $P_\pi A = PAP^T$ for some permutation matrix $P$:

**Lemma C.1.** $A = MNM^T \iff P_\pi A = (P_\pi M)(P_\pi N)(P_\pi M)^T$.

Recall the modified Cholesky decomposition of $A$ (also called the LDLT decomposition): $A = LDL^T$ for a lower triangular matrix $L$, with unit diagonal entries, and a diagonal matrix $D$. When $A$ is positive definite, the pair $(L, D)$ is unique and we refer to it as the *Cholesky decomposition of $A$*.

Let us denote the set of all pairs $(\widetilde{B}, \widetilde{\Omega})$ satisfying $\Sigma^{-1} = (I - \widetilde{B})\widetilde{\Omega}^{-1}(I - \widetilde{B}^T)$ (equivalently, (3)) as $\mathfrak{D}'$. Next, note that $\widetilde{B} \in \mathbb{D}$ if and only if $P_\pi \widetilde{B}$ is lower triangular for some permutation $\pi$. Lemma C.1 implies that $(\widetilde{B}, \widetilde{\Omega}) \in \mathfrak{D}'$ iff $(I - P_\pi \widetilde{B}, P_\pi \widetilde{\Omega}^{-1})$ is a Cholesky decomposition of $P_\pi \Sigma^{-1}$ for some $\pi$.

Now, $(I - P_\pi \widetilde{B}(\pi), P_\pi \widetilde{\Omega}(\pi)^{-1})$ is also a Cholesky decomposition of $P_\pi \Sigma^{-1}$. Since the Cholesky decomposition is unique for positive definite matrices, we have $(\widetilde{B}, \widetilde{\Omega}) \in \mathfrak{D}'$ iff $(\widetilde{B}, \widetilde{\Omega}) = (\widetilde{B}(\pi), \widetilde{\Omega}(\pi))$ for some $\pi$, which gives the desired result, since $\mathfrak{D}(\Sigma)$ is the projection of $\mathfrak{D}'$ onto its first coordinate.

## C.3 Proof of Lemma B.1

The first conclusion (a) follows from elementary properties of conditional expectation and the identity

$$\mathbb{E}(X_j \mid X_{S_j(\pi)}) = \widetilde{\beta}_j(\pi)^T X.$$

To prove (b), fix $\widehat{\pi} \in \widehat{\Pi}$ and let $S_j = S_j(\widehat{\pi})$. If $\widehat{\beta}_j \in \widehat{\Theta}_\lambda(\mathbf{x}_j, \mathbf{X}; S_j)$ for each $j$, then evidently $\widehat{B} = [\widehat{\beta}_1 \mid \cdots \mid \widehat{\beta}_p]$ minimizes $Q(B)$ over $\mathbb{D}[\widehat{\pi}]$ (cf. (18)). For the reverse direction, recall that $\mathbf{X}_{S_j}$ is the $n \times |S_j|$ matrix formed by extracting the columns in $S_j$, and similarly for $(\beta_j)_{S_j}$. For any $B \in \mathbb{D}[\pi]$ we have $(\beta_j)_{S_j^c} = 0$ for each $j$, so we can write fix $\widehat{\pi} \in \widehat{\Pi}$ and let $S_j = S_j(\widehat{\pi})$. If $\widehat{\beta}_j \in \widehat{\Theta}_\lambda(\mathbf{x}_j, \mathbf{X}; S_j)$ for each $j$, then evidently $\widehat{B} = [\widehat{\beta}_1 \mid \cdots \mid \widehat{\beta}_p]$ minimizes $Q(B)$ over $\mathbb{D}[\widehat{\pi}]$. For the reverse direction, recall that $\mathbf{X}_{S_j}$ is the $n \times |S_j|$ matrix formed by extracting the columns in $S_j$, and similarly for $(\beta_j)_{S_j}$. For any $B \in \mathbb{D}[\pi]$ we have $(\beta_j)_{S_j^c} = 0$ for each $j$, so we can write

$$\frac{1}{2n}\|\mathbf{X} - \mathbf{X}B\|_F^2 + \rho_\lambda(B) = \sum_{j=1}^p \left\{ \frac{1}{2n}\|\mathbf{x}_j - \mathbf{X}\beta_j\|_2^2 + \rho_\lambda(\beta_j) \right\}$$

$$= \sum_{j=1}^p \left\{ \frac{1}{2n}\|\mathbf{x}_j - \mathbf{X}_{S_j}(\beta_j)_{S_j}\|_2^2 + \rho_\lambda((\beta_j)_{S_j}) \right\}.$$

Then $\widehat{B} \in \min_{\mathbb{D}[\pi]} Q(B)$ if and only if

$$\widehat{\beta}_j \in \arg\min_\beta \frac{1}{2n}\|\mathbf{x}_j - \mathbf{X}\beta\|_2^2 + \rho_\lambda(\beta) \quad \text{subject to } \beta_{S_j^c} = 0.$$

In other words, $\widehat{\beta}_j \in \widehat{\Theta}_\lambda(\mathbf{x}_j, \mathbf{X}; S_j)$ for each $j$. Since $\widehat{\pi} \in \widehat{\Pi}$ was arbitrary, the desired claim follows.

## C.4 Proof of Lemma B.2

The proof relies on the following property of $L^2$ projections: For any two sets $S, R \subset [p]_j$, we have

$$\beta_j(S \cup R) = \beta_j(S) \iff \varepsilon_j(S) \perp\!\!\!\perp X_i, \ \forall i \in R. \tag{42}$$

To lighten the notation, let $S^* = m_j(S)$. Note that $\beta_j(S) = \beta_j(S^*)$ since $\mathrm{supp}(\beta_j(S)) = S^*$. It follows from (42) that $\varepsilon_j(S^*) \perp\!\!\!\perp X_i$ for $i \in S \setminus S^*$. Similarly, since $\mathrm{supp}(\beta_j(T_k)) = S^*$, we have $\varepsilon_j(S^*) \perp\!\!\!\perp X_i$ for $i \in T_k \setminus S^*$ and $k = 1, 2$. It follows that

$$\varepsilon_j(S^*) \perp\!\!\!\perp X_i, \ \forall i \in (T_1 \setminus S^*) \cup (T_2 \setminus S^*)$$

hence the application of (42) in the reverse direction yields

$$\beta_j(T_1 \cup T_2) = \beta_j\big(S^* \cup (T_1 \setminus S^*) \cup (T_2 \setminus S^*)\big) = \beta_j(S^*) = \beta_j(S).$$

## C.5 Proof of Lemma B.3

It suffices to show
$$A(Z, \theta^*; U)^c \subset A(Z, \theta^*; S)^c.$$

Suppose $w \in A(Z, \theta^*; U)^c$, i.e., $\mathrm{supp}(\widetilde{\theta}) = \mathrm{supp}(\theta^*) := S^*$ for any $\widetilde{\theta} \in \widehat{\Theta}_\lambda(Z\theta^* + w, Z; U)$. We wish to show that for any $\widehat{\theta} \in \widehat{\Theta}_\lambda(Z\theta^* + w, Z; S)$, it must also be true that $\mathrm{supp}(\widehat{\theta}) = S^*$. Let

$$F(\theta) = \frac{1}{2n} \|Z(\theta^* - \theta) + w\|_2^2 + \rho_\lambda(\theta)$$

denote the objective function in Definition A.4 of $\widehat{\Theta}_\lambda(y, Z; S)$ with $y = Z\theta^* + w$. Since $\mathrm{supp}(\widehat{\theta}) \subset S \subset U$, $\widehat{\theta}$ is feasible for the $U$-restricted problem, whence

$$F(\widetilde{\theta}) \le F(\widehat{\theta})$$

for any $\widetilde{\theta} \in \widehat{\Theta}_\lambda(Z\theta^* + w, Z; U)$. But $\widetilde{\theta}$ is also feasible for the $S$-restricted problem since $\mathrm{supp}(\widetilde{\theta}) = S^* \subset S$, so that

$$F(\widetilde{\theta}) \ge F(\widehat{\theta}) \implies F(\widetilde{\theta}) = F(\widehat{\theta}).$$

Since the value $F(\widetilde{\theta})$ is by definition the global minimum of $F$ for the $U$-restricted problem and $\mathrm{supp}(\widehat{\theta}) \subset U$, $\widehat{\theta}$ must be a global minimizer of $F$ for the $U$-restricted problem, i.e., $\widehat{\theta} \in \widehat{\Theta}_\lambda(Z\theta^* + w, Z; U)$, whence $\mathrm{supp}(\widehat{\theta}) = S^*$ as desired.

## C.6 Proof of Corollary B.4

By Lemma B.3 and the fact that $S \subset M_j(S)$, we have

$$\mathcal{A}\Big(\widetilde{\varepsilon}_j(S), \mathbf{X}, \beta_j(S); S\Big) \subset \mathcal{A}\Big(\widetilde{\varepsilon}_j(S), \mathbf{X}, \beta_j(S); M_j(S)\Big). \tag{43}$$

Using (26) and (27), we have the following identity:

$$\mathcal{A}\Big(\widetilde{\varepsilon}_j(S), \mathbf{X}, \beta_j(S); M_j(S)\Big) = \mathcal{A}\Big(\widetilde{\varepsilon}_j(M_j(S)), \mathbf{X}, \beta_j(M_j(S)); M_j(S)\Big).$$

Plugging this into (43) yields the desired result.

## C.7 Proof of Proposition B.5

Throughout, for simplicity, let

$$\mathcal{A}_S := \mathcal{A}(\widetilde{\varepsilon}_j(S), \mathbf{X}, \beta_j(S); S).$$

Fix $S \subset [p]_j$ and let $\theta^* = \beta_j(S)$, $s^* = |m_j(S)| = \|\theta^*\|_0$ and $\varepsilon^* = \widetilde{\varepsilon}_j(S)$ so that $\mathcal{A}_S = \mathcal{A}(\varepsilon^*, \mathbf{X}, \theta^*; S)$. Note that $\mathcal{A}(\varepsilon^*, \mathbf{X}, \theta^*; S)$ represents the following model selection failure:

$$\mathrm{supp}(\widehat{\theta}) \ne \mathrm{supp}(\theta^*) \quad \exists \widehat{\theta} \in \widehat{\Theta}_\lambda(\mathbf{X}\theta^* + \varepsilon^*, \mathbf{X}; S).$$

Since $\mathrm{supp}(\theta^*) \subset S$, we can restrict $\mathbf{X}$ and $\theta^*$ to $S$, so that the above is equivalent to

$$\mathrm{supp}(\widehat{\theta}) \ne \mathrm{supp}(\theta_S^*) \quad \exists \widehat{\theta} \in \widehat{\Theta}_\lambda(\mathbf{X}_S \theta_S^* + \varepsilon^*, \mathbf{X}_S).$$

which is the same event as $\mathcal{A}(\varepsilon^*, \mathbf{X}_S, \theta_S^*)$. To summarize, $\mathcal{A}_S = \mathcal{A}(\varepsilon^*, \mathbf{X}_S, \theta_S^*)$.

Since $\varepsilon^*$ is independent of $\mathbf{X}_S$ by Lemma B.1(a), by conditioning on $\mathbf{X}_S$ we are dealing with a fixed design regression problem with Gaussian noise $\varepsilon^* = \widetilde{\varepsilon}_j(S) \sim \mathcal{N}_n(0, \omega_j^2(S)I_n)$. We obtain

$$\begin{aligned}
\mathbb{P}(\mathcal{A}_S) &= \mathbb{E}\Big[\mathbb{P}\big(\mathcal{A}(\varepsilon^*, \mathbf{X}_S, \theta_S^*)\big) \mid \mathbf{X}_S\Big] \\
&\le \mathbb{E}\exp[-\Phi_\lambda(\mathbf{X}_S, \theta_S^*, \omega_j^2(S))] \\
&= \mathbb{E}\exp(-\Phi_j(S)),
\end{aligned} \tag{44}$$

where the last line uses (31). Now we have

$$\left\{ \operatorname{supp}(\widehat{\beta}_j(T)) \neq \operatorname{supp}(\beta_j(T)), \exists\, T \in \mathcal{T}_j(S) \right\} = \bigcup_{T \in \mathcal{T}_j(S)} \mathcal{A}_T = \mathcal{A}_{M_j(S)}, \qquad (45)$$

where the first equality is by (28) and the second follows from Corollary B.4. Note that this is the key step where the reduction occurs. Hence, combining (45) with (44) we have

$$\begin{aligned}
\mathbb{P}\Big( \bigcup_{S \subset [p]_j} \mathcal{A}_S \Big) &= \mathbb{P}\Big( \bigcup_{S \subset [p]_j} \mathcal{A}_{M_j(S)} \Big) \\
&= \mathbb{P}\Big( \bigcup_{T \in m_j(\Sigma)} \mathcal{A}_{M_j(T)} \Big) \\
&\leq \sum_{T \in m_j(\Sigma)} \mathbb{P}(\mathcal{A}_{M_j(T)}) \leq \sum_{T \in m_j(\Sigma)} \mathbb{E}\exp(-\Phi_j(M_j(T))),
\end{aligned}$$

which is the desired probability bound.

## C.8   Proof of Proposition B.7

We work with the column decomposition of the loss

$$\ell_j(\beta) := \frac{1}{n}\|\mathbf{X}(e_j - \beta)\|_2^2,$$

$$\mathbb{E}\ell_j(\beta) = \frac{1}{2}(e_j - \beta)^T \Sigma (e_j - \beta),$$

where $e_j \in \mathbb{R}^p$ is the $j$th standard basis vector. The overall loss can be written as

$$\ell(B) = \sum_{j=1}^n \ell_j(\beta_j)$$

where $\beta_j$ is the $j$th column of $B$. Let us also define so that

$$J(\beta) = \frac{1}{n}\|\mathbf{X}\beta\|_2^2, \quad \text{and,} \quad \mathbb{E}J(\beta) := \beta^T \Sigma \beta$$

so that $\ell_j(\beta) = \frac{1}{2}J(e_j - \beta)$ and $\mathbb{E}\ell_j(\beta) = \frac{1}{2}\mathbb{E}J(e_j - \beta)$. It is easier to work with $J$. Let $\mathbf{X}_i^T$ be the $i$th row of $\mathbf{X}$. Then, $J(\beta) = \frac{1}{2n}\sum_{i=1}^n (\mathbf{X}_i^T \beta)^2$.

Let $K = \mathbb{D}_G$ and $K_j$ denote the set of $\beta_j$ for $B \in K$. Define

$$\mathbb{B}_0^{-j}(s) = \{x \in \mathbb{R}^p : \|x\| \leq s,\ x_j = 0\}. \qquad (46)$$

Note that $\beta_j \in \mathbb{B}_0^{-j}(s)$ for every $B \in \mathbb{D}_G$ and in particular $\widetilde{\beta}_j(\pi) \in \mathbb{B}_0^{-j}(s)$ if $\widetilde{B}(\pi) \in \mathbb{D}_G$. Finally, define

$$\kappa(\Sigma; s) := \frac{\|\Sigma\|_{(2s+2)}}{r_{\min}^{(s+1)}(\Sigma)} \qquad (47)$$

where

$$\|\Sigma\|_{(s)} := \max_{S:|S|=s} \|\Sigma_S\|, \qquad (48)$$

$$r_{\min}^{(s)}(\Sigma) := \inf_{S:|S|=s} r_{\min}(\Sigma_S). \qquad (49)$$

*Proof of Proposition B.7.* For any $\beta \in K_j$, we have

$$\|e_j - \beta\|^2 = 1 + \|\beta\|^2. \qquad (50)$$

For any $t \leq 1$, applying Lemma C.5 we have that on the event $\mathcal{B}_{2s}$ (defined in (53))

$$|\ell_j(\beta) - \mathbb{E}\ell_j(\beta)| \leq t\, J(\beta) + 3\gamma_2\left(1 + \|\beta\|_2^2\right)\varepsilon\,\|\Sigma\|_{(2s)} \text{ for all } \beta \in K_j \qquad (51)$$

fails with probability at most $2\left(\frac{3ep}{s\varepsilon}\right)^{s}e^{-cnt^{2}}$. A further union bound over $j = 1, \ldots, p$ gives that

$$|\ell_j(\beta) - \mathbb{E}\ell_j(\beta)| \le t\, J(\beta) + 3\gamma_2\left(1 + \|\beta\|_2^2\right)\varepsilon\,\|\Sigma\|_{(2s)}, \tag{52}$$
$$\text{for all } \beta \in K_j \text{ and all } j \in [p]$$

fails with probability at most

$$2p\left(\frac{3ep}{s\varepsilon}\right)^{s}e^{-cnt^{2}} + \mathbb{P}(\mathcal{B}_{2s}^c) \ \le\ 2p\left(\frac{3ep}{s\varepsilon}\right)^{s}e^{-cnt^{2}} + \left(\frac{ep}{2s}\right)^{-c_1 2s} =: T_1 + T_2,$$

where we invoked Lemma C.4 to bound $\mathbb{P}(\mathcal{B}_{2s}^c)$. Take $t = \varepsilon$ and let $N = p^{1/s}3ep/s$ so that the first term in the bound is

$$T_1 := 2\left(\frac{N}{\varepsilon}\right)^{s}e^{-cn\varepsilon^2}.$$

Take $\varepsilon$

$$\varepsilon^2 = \frac{2}{c}\frac{s}{n}\log N \le 1$$

which gives the following bound,

$$T_1 \le 2\left(\varepsilon N\right)^{-s} = 2p^{-1}\left(\gamma_1\frac{3ep}{s}\right)^{-s}.$$

where we note that with our choices, we have $t = \varepsilon = \gamma_1$ as defined in (8).

Note that for any $\beta \in K_j$, $(1 + \|\beta\|^2)\, r_{\min}^{(s+1)}(\Sigma) \le 2\ell_j(\beta)$, which implies $1 + \|\beta\|^2 \le 2\ell_j(\beta)/r_{\min}^{(s+1)}(\Sigma)$. Plugging this upper bound into (52) and summing over $j$ gives (34), where $\kappa(\Sigma; s)$ is defined by (47). The proof is complete. $\qquad\square$

Below we prove the various technical lemmas required in the previous proof.

**Lemma C.2.** *We have*

$$\mathbb{P}\left(|J(\beta) - \mathbb{E}J(\beta)| \ge t \cdot J(\beta)\right) \le 2\exp\left[-\tfrac{1}{8}\,n\cdot\min(t^2, t)\right], \quad t \ge 0.$$

*Proof.* Note that $\mathbf{X}_i^T\beta/\sqrt{\mathbb{E}J(\beta)} \sim N(0,1)$ iid for each $i = 1, \ldots, n$. Then,

$$\frac{J(\beta)}{\mathbb{E}J(\beta)} = \frac{1}{n}\sum_{i=1}^{n}\left(\frac{\mathbf{X}_i^T\beta}{\sqrt{\mathbb{E}J(\beta)}}\right)^2 \sim \chi_n^2$$

and the claim follows from $\chi^2$ concentration. $\qquad\square$

Let $\mathcal{B}_{2s}$ be the following event:

$$\mathcal{B}_{2s} := \left\{g_{n,s}(\varepsilon) \le \varepsilon\sqrt{\gamma_2}\|\Sigma\|_{(2s)}^{1/2}, \quad \forall\varepsilon > 0\right\}. \tag{53}$$

where

$$g_{n,s}(\varepsilon) = \sup_{\|u\|_2 \lesssim \varepsilon,\ \|u\|_0 \lesssim s}\frac{1}{\sqrt{n}}\|\mathbf{X}u\|_2, \quad\text{and} \tag{54}$$
$$\|\Sigma\|_{(s)} := \max_{S:|S|=s}\|\Sigma_S\|. \tag{55}$$

**Lemma C.3.** *For any $c_1 > 0$, with probability at least $1 - (ep/s)^{-c_1 s}e^{-u^2/2}$,*

$$g_{n,s}(\varepsilon) \le \varepsilon\|\Sigma\|_{(s)}^{1/2}\left(1 + C\sqrt{\frac{s\log(ep/s)}{n}} + \frac{u}{\sqrt{n}}\right), \quad \forall\varepsilon > 0.$$

*where $C = \sqrt{2(c_1 + 1)} + 1$.*

*Proof.* Note that $g_{n,s}(\varepsilon) = \varepsilon\, g_{n,s}(1)$ which is obtained by the change of variable $u \to \varepsilon u$. We have

$$g_{n,s}(1) = \max_{S:|S|=s} \sup_{\|u\|_2 \leq 1} \frac{1}{\sqrt{n}} \|\mathbf{X}_S u\|_2.$$

Let $W_S = \mathbf{X}_S \Sigma_S^{-1/2}$ so that $W_S \sim N(0, I_s)$. Then

$$\mathbb{P}\left(\|W_S\| > \sqrt{s} + \sqrt{n} + t\right) \leq \exp(-t^2/2).$$

We also have $\|\mathbf{X}_S u\|_2 = \|\Sigma_S^{1/2} W_S u\|_2 = \|\Sigma_S^{1/2}\|\|W_S\|\|u\|_2$. Thus,

$$g_{n,s}(1) \leq \max_{S:|S|=s} \|\Sigma_S^{1/2}\| \cdot \max_{S:|S|=s} \frac{1}{\sqrt{n}} \|W_S\|$$

and hence

$$\mathbb{P}\left(\max_{S:|S|=s} \frac{1}{\sqrt{n}} \|W_S\| > \sqrt{\frac{s}{n}} + 1 + \frac{t}{\sqrt{n}}\right) \leq \binom{p}{s} \exp(-t^2/2)$$

Taking $t = \sqrt{2(c_1+1)s \log(ep/s)} + u$, the above probability is bounded by

$$\binom{p}{s} e^{-t^2/2} \leq (ep/s)^s (ep/s)^{-(c_1+1)s} e^{-u^2/2} = (ep/s)^{-c_1 s} e^{-u^2/2}$$

Letting $C = \sqrt{2(c_1+1)} + 1$ and noting that $\|A^{1/2}\| = \|A\|^{1/2}$, the result follows. $\qquad\square$

**Lemma C.4.** *Let $\mathcal{B}_{2s}$ be defined as in* (53). *Then* $\mathbb{P}(\mathcal{B}_{2s}^c) \leq (ep/(2s))^{-c_1 2s}$.

*Proof.* Apply Lemma C.3 with $s$ replaced with $2s$ and $u = 0$, and note that

$$2s \log(ep/(2s)) = 2s[\log(ep/s) - \log 2] \leq 2s \log(ep/s),$$

we observe that $\mathcal{B}_{2s}$ fails with probability at most $(ep/(2s))^{-c_1 2s}$. $\qquad\square$

Define the $L^q$ balls

$$\mathbb{B}_q(r) := \{x \in \mathbb{R}^p : \|x\|_q \leq r\}.$$

**Lemma C.5.** *For any $t \leq 1$: On the event $\mathcal{B}_{2s}$,*

$$|J(\beta) - \mathbb{E}J(\beta)| \leq t\, J(\beta) + 3\gamma_2 \|\beta\|_2^2\, \varepsilon \|\Sigma\|_{(2s)}, \quad \text{for all } \beta \in \mathbb{B}_0(s) \tag{56}$$

*fails with probability at most* $2\left(\frac{3ep}{s\varepsilon}\right)^s e^{-cnt^2}$.

*Proof.* Write (56) in the form $g(\beta) \leq 0$ and observe that $g$ is homogeneous of order two: $g(r\beta) = r^2 g(\beta)$ for any $r \in \mathbb{R}$ and $\beta \in \mathbb{R}^p$. Thus, it is enough to establish the bound for $\beta \in \mathbb{B}_2(1)$. The general case is then obtained by applying the bound to $\beta/\|\beta\|_2$.

Let $N_j \subseteq \mathbb{B}_0(s) \cap \mathbb{B}_2(1)$ be an $\varepsilon$-net for $\mathbb{B}_0(s) \cap \mathbb{B}_2(1)$ in $\ell_2$ norm. Then, $|N_j| \leq \binom{p}{s}(3/\varepsilon)^s$. For any $\beta, \beta' \in \mathbb{B}_0(s) \cap \mathbb{B}_2(1)$, using

$$\left|\|x\|^2 - \|y\|^2\right| \leq \left|\|x\| - \|y\|\right|(\|x\| + \|y\|) \leq \|x - y\|\|x + y\|$$

we have on event $\mathcal{B}_{2s}$,

$$\begin{aligned}
\left|J(\beta) - J(\beta')\right| &= \left|\frac{1}{n}\|\mathbf{X}\beta\|_2^2 - \frac{1}{n}\|\mathbf{X}\beta'\|_2^2\right| \\
&\leq \frac{1}{n}\|\mathbf{X}(\beta - \beta')\|_2 \cdot \|\mathbf{X}(\beta + \beta')\|_2 \\
&\leq g_{n,2s}(\|\beta - \beta'\|_2) \cdot g_{n,2s}(\|\beta + \beta'\|_2) \\
&\leq \sqrt{\gamma_2}\|\Sigma\|_{(2s)}^{1/2}\|\beta - \beta'\|_2 \cdot \sqrt{\gamma_2}\|\Sigma\|_{(2s)}^{1/2}\|\beta + \beta'\|_2 \\
&\leq 2\gamma_2\|\Sigma\|_{(2s)}\|\beta - \beta'\|_2,
\end{aligned}$$

where we have used $\|\beta + \beta'\|_2 \leq \|\beta\| + \|\beta'\| \leq 2$. A similar bound holds for the expectation $\mathbb{E}J(\beta)$, i.e.

$$
\begin{aligned}
\left| \mathbb{E}J_j(\beta) - \mathbb{E}J_j(\beta') \right| &= \left| \|\Sigma^{1/2}\beta\|_2^2 - \|\Sigma^{1/2}\beta'\|_2^2 \right| \\
&\leq \|\Sigma^{1/2}(\beta - \beta')\|_2 \cdot \|\Sigma^{1/2}(\beta + \beta')\|_2 \\
&\leq \|\Sigma\|_{(2s)}^{1/2}\|\beta - \beta'\|_2 \cdot \|\Sigma\|_{(2s)}^{1/2}\|\beta + \beta'\|_2 \\
&\leq 2\|\Sigma\|_{(2s)}\|\beta - \beta'\|_2.
\end{aligned}
$$

It follows that for any $f(\cdot)$, with $\gamma = 2(\gamma_2 + 1)$,

$$
\begin{aligned}
\sup_{\beta \,\in\, \mathbb{B}_0(s) \cap \mathbb{B}_2(1)} & \left( |J(\beta) - \mathbb{E}J(\beta)| - f(\beta) \right) \\
&\leq \gamma\|\Sigma\|_{(2s)}\,\varepsilon + \sup_{\beta \,\in\, N_j} \left( |J(\beta) - \mathbb{E}J(\beta)| - f(\beta) \right).
\end{aligned}
\tag{57}
$$

We now have, for any $t \geq 0$,

$$
\begin{aligned}
\mathbb{P}\left( \left\{ \sup_{\beta \,\in\, \mathbb{B}_0(s) \cap \mathbb{B}_2(1)} \left[ |J(\beta) - \mathbb{E}J(\beta)| - t\mathbb{E}J(\beta) \right] \geq \gamma\|\Sigma\|_{(2s)}\varepsilon \right\} \cap \mathcal{B}_{2s} \right) \\
\leq \mathbb{P}\left( \left\{ \sup_{\beta \,\in\, N_j} \left[ |J(\beta) - \mathbb{E}J(\beta)| - t\mathbb{E}J(\beta) \right] \geq 0 \right\} \cap \mathcal{B}_{2s} \right) \\
\leq 2|N_j| \exp\left[ -c\,n \cdot \min(t^2, t) \right].
\end{aligned}
$$

Noting that $|N_j| \leq \left( \frac{ep}{s} \right)^s \left( \frac{3}{\varepsilon} \right)^s$ and $\gamma \leq 3\gamma_2$ completes the proof. $\qquad\square$

## C.9 Proof of Lemma B.8

We will need the following lemma, whose proof is a straightforward calculation:

**Lemma C.6.** *Let $D$ be a diagonal matrix and $A = (a_{ij}) = (I - L)(I - L)^T$ where $L$ is a strictly lower triangular matrix. Then $\mathrm{tr}(AD) \geq \mathrm{tr}(D)$ with equality if and only if $A = I$ (i.e. $L = 0$).*

We now prove Lemma B.8. Write $P$ for the permutation matrix corresponding to $\pi$. Now suppose that $B \in \mathbb{D}_p[\pi]$, so that $PBP^T = L$ and $P\widetilde{B}(\pi)P^T = \widetilde{L}$ are strictly lower triangular matrices. Then

$$
\begin{aligned}
\mathbb{E}\left[ \ell(B) \right] &= \tfrac{1}{2}\, \mathrm{tr}\left[ (I - \widetilde{B}(\pi))^{-1}(I - B)(I - B)^T(I - \widetilde{B}(\pi))^{-T}\widetilde{\Omega}(\pi) \right] \\
&= \tfrac{1}{2}\, \mathrm{tr}\left[ P(I - \widetilde{B}(\pi))^{-1}(I - B)(I - B)^T(I - \widetilde{B}(\pi))^{-T}\widetilde{\Omega}(\pi)P^T \right] \\
&= \tfrac{1}{2}\, \mathrm{tr}\left[ (I - \widetilde{L})^{-1}(I - L)(I - L)^T(I - \widetilde{L})^{-T}P\widetilde{\Omega}(\pi)P^T \right].
\end{aligned}
$$

Note that $(I - \widetilde{L})^{-1}(I - L)$ is lower triangular, so that $(I - \widetilde{L})^{-1}(I - L)(I - L)^T(I - \widetilde{L})^{-T} := A$ is of the form $A = (I - \overline{L})(I - \overline{L})^T$ for some strictly lower triangular matrix $\overline{L}$. In particular, restricted to $\mathbb{D}_p[\pi]$, $\mathbb{E}\left[ \ell(B) \right]$ is of the form $\mathrm{tr}(AD)$ for the diagonal matrix $D := P\widetilde{\Omega}(\pi)P^T$. Inequality (36) then follows from Lemma C.6.

Finally, if there is equality in (36), then Lemma C.6 implies that $(I - \widetilde{L})^{-1}(I - L)(I - L)^T(I - \widetilde{L})^{-T} = I$, or

$$
(I - L)(I - L)^T = (I - \widetilde{L})^T(I - \widetilde{L}),
$$

and the desired claim follows from the uniqueness of the Cholesky decomposition.

## C.10 Proof of Lemma B.9

Observe that for any $\pi \in \mathbb{S}_p$,

$$
Q(\widehat{B}) \leq Q(\widetilde{B}(\pi)).
\tag{58}
$$

Moreover, we have the following alternative expression for $Q$:

$$Q(B) = \frac{1}{2n}\|\mathbf{X}(\widetilde{B}(\widehat{\pi}) - B) + \mathbf{E}(\widehat{\pi})\|_F^2 + \rho_\lambda(B), \quad \text{for any } \widehat{\pi} \in \widehat{\Pi}. \tag{59}$$

Thus, using (58) and (59),

$$\begin{aligned}
0 &\leq Q(\widetilde{B}(\pi)) - Q(\widehat{B}) \\
&= \frac{1}{2n}\|\mathbf{E}(\pi)\|_F^2 - \frac{1}{2n}\|\mathbf{X}(\widetilde{B}(\widehat{\pi}) - \widehat{B}) - \mathbf{E}(\widehat{\pi})\|_F^2 + \rho_\lambda(\widetilde{B}(\pi)) - \rho_\lambda(\widehat{B}) \\
&= \frac{1}{2n}\|\mathbf{E}(\pi)\|_F^2 - \frac{1}{2n}\|\mathbf{E}(\widehat{\pi})\|_F^2 - \frac{1}{2n}\|\mathbf{X}(\widetilde{B}(\widehat{\pi}) - \widehat{B})\|_F^2 \\
&\quad + \frac{1}{n}\operatorname{tr}\left(\mathbf{E}(\widehat{\pi})^T\mathbf{X}(\widetilde{B}(\widehat{\pi}) - \widehat{B})\right) + \rho_\lambda(\widetilde{B}(\pi)) - \rho_\lambda(\widehat{B}).
\end{aligned}$$

Since (58) holds for any $\pi$, this completes the proof.

# D  Auxiliary results

This section provides some additional results which are needed to prove Proposition B.10. This involves several steps: 1) Bounding the estimation error $\|\widehat{B} - \widetilde{B}(\widehat{\pi})\|_r$ (Section D.1), 2) Controlling the terms (B.9b) and (B.9c) (Section D.2), and 3) Controlling (B.9a), which invokes the minimum-trace permutations $\Pi_0$ (Section D.3). After dealing with these prerequisites, we prove Proposition B.10 in Appendix D.4.

For any $\delta \in (0, 1)$, $\lambda \geq 0$, $\delta_0 > 0$, and $\pi \in \mathbb{S}_p$, define the following event:

$$\mathcal{G}(\delta_0, \lambda; \pi) = \left\{ \frac{1}{2n}\|\mathbf{E}(\pi)\|_F^2 - \frac{1}{2n}\|\mathbf{E}(\widehat{\pi})\|_F^2 \leq \delta_0 \rho_\lambda(\widetilde{B}(\pi)) \right\}. \tag{60}$$

The idea is to show that on this event (along with (74) and (75)), the desired conclusions hold. In Appendix D.3, we provide an explicit bound on the probability of $\mathcal{G}(\delta_0, \lambda; \pi)$.

## D.1  Uniform deviation bounds

The purpose of this section is to control the estimation error $\|\widehat{B} - \widetilde{B}(\widehat{\pi})\|_r$ via Proposition D.2, which is needed in the proof of Lemma D.7. This lemma—which is also proved in this Appendix—is a key prerequisite in the proof of Proposition B.10.

We start by establishing a general bound on the $\ell_r$ ($r = 1, 2$) estimation errors for a fixed design regression problem with a general regularizer $\rho_\lambda$. The objective here is to derive conditions under which we can guarantee such bounds for a fixed design problem, and then show that these conditions hold uniformly for all neighbourhood problems. The conditions we will need are familiar from the literature: A *Gaussian width condition* and a *restricted eigenvalue condition*.

For the rest of this subsection, we let $Z \in \mathbb{R}^{n \times m}$ and $w \in \mathbb{R}^n$ be a fixed matrix and fixed vector, respectively.

*Definition* D.1 (Gaussian width). We say that the *Gaussian width (GW) condition* holds for $(w, Z)$ relative to $\rho_\lambda$ if there is a numerical constant $\delta \in (0, 1)$ such that

$$\frac{1}{n}|\langle w, Zu \rangle| \leq \delta\left[\frac{1}{2n}\|Zu\|_2^2 + \rho_\lambda(u)\right], \ \forall u \in \mathbb{R}^m,$$

in which case we write $(w, Z) \in \mathrm{GW}_{\rho_\lambda}(\delta)$. If this inequality is strict for all $u \neq 0$, we write $(w, Z) \in \mathrm{GW}_{\rho_\lambda}^\circ(\delta)$.

We will be interested in the case where both $w$ and $Z$ are allowed to be random but independent. In this setting, for Gaussian designs considered in this paper, the GW condition holds with high probability for the $\ell_1$ penalty (this follows from a standard Hölder inequality argument), and has

similarly been shown to hold for penalties induced by $\ell_q$ norms for $0 \leq q \leq 1$ [49]. Zhang and Zhang [74] provide a version of this condition that applies to general nonconvex regularizers.

Before we proceed, let us note the following key relation between model selection consistency and the GW condition:

**Lemma D.1.** *Consider the setup of Lemma B.3, namely, the regression problem* $\mathbf{y} = Z\theta^* + \mathbf{w}$ *but with* $\theta^* = 0$. *Then*

$$\mathcal{A}(\mathbf{w}/\delta, Z, 0)^c = \left\{ (\mathbf{w}, Z) \in \mathrm{GW}^\circ_{\rho_\lambda}(\delta) \right\}.$$

*Proof.* If $(\mathbf{w}, Z) \in \mathrm{GW}^\circ_{\rho_\lambda}(\delta)$, then for any $u \neq 0$,

$$\frac{\delta}{2n}\|Zu\|_2^2 - \frac{1}{n}\mathbf{w}^T Zu + \delta\rho_\lambda(u) > 0$$
$$\iff \frac{1}{2n}\|\mathbf{w}/\delta - Zu\|_2^2 + \rho_\lambda(u) > \frac{1}{2n}\|\mathbf{w}/\delta\|_2^2.$$

The latter inequality implies

$$\{0\} = \arg\min_u \|\mathbf{w}/\delta - Zu\|_2^2/(2n) + \rho_\lambda(u),$$

that is, $0$ is the unique global minimizer of the right hand side. Recalling the definition of $\mathcal{A}(\mathbf{w}/\delta, Z, 0)$ in (22), we obtain the desired result. $\qquad\square$

Thus, in order to ensure the GW condition for $(\mathbf{w}, Z_S)$, it suffices to show that the corresponding regression problem is model selection consistent when the *true coefficients are all set to zero* and the noise variance is inflated by a factor of $1/\delta^2$. [74] refer to this property as *null-consistency*.

For any set $A \subset [m]$ and $\xi > 0$, define the following "cone":

$$C_{\rho_\lambda}(A, \xi) := \{u \in \mathbb{R}^m : \rho_\lambda(u_{A^c}) \leq \xi\rho_\lambda(u_A)\}. \tag{61}$$

This definition also depends on the ambient dimension $m$; when we wish to emphasize this we will write $C^m_{\rho_\lambda}(A, \xi)$. The term "cone" here is used in an extended sense, in analogy with the $\ell_1$ cone found in previous work.

*Definition* D.2 (Generalized restricted eigenvalue). The *generalized restricted eigenvalue (RE) constant* of $Z$ with respect to $\rho_\lambda$ over a subset $A$ is

$$\phi^2_{\rho_\lambda}(Z, A; \xi) := \inf \left\{ \frac{\|Zu\|_2^2}{n\|u\|_2^2} : u \in C_{\rho_\lambda}(A, \xi), \, u \neq 0 \right\}. \tag{62}$$

In the sequel, we often suppress the dependence of the generalized RE constants on $\lambda$ and $\xi$, writing $\phi^2_\rho(Z, A) = \phi^2_{\rho_\lambda}(Z, A; \xi)$. Note that the usual restricted eigenvalue is equivalent to the special case $\rho_\lambda = \lambda\|\cdot\|_1$ [3].

Consider the usual linear regression set up, $y = Z\theta^* + w$, where $\theta^* \in \mathbb{R}^m$ and we define $S^* = \mathrm{supp}(\theta^*)$. The following general result establishes that the two conditions $(w, Z) \in \mathrm{GW}_\rho(\delta)$ and $\phi^2_\rho(Z, S^*) > 0$ are sufficient to bound the deviation $\widehat{\theta} - \theta^*$:

**Theorem D.1.** *Assume* $(w, Z) \in \mathrm{GW}_{\rho_\lambda}(\delta)$ *for some* $\rho_\lambda$ *satisfying Condition A.1 and* $\delta \in (0, 1)$. *Let* $\xi = \xi(\delta) := (1 + \delta)/(1 - \delta)$ *and assume* $\phi^2 := \phi^2_\rho(Z, S^*; \xi) > 0$. *Then any* $\widehat{\theta} \in \widehat{\Theta}_\lambda(Z\theta^* + w, Z)$ *satisfies*

$$\|\widehat{\theta} - \theta^*\|_2 \leq C_2(\rho_\lambda, \xi, \phi) \cdot \|\theta^*\|_0^{1/2}, \tag{63}$$
$$\|\widehat{\theta} - \theta^*\|_1 \leq C_1(\rho_\lambda, \xi, \phi) \cdot \|\theta^*\|_0. \tag{64}$$

*Remark* D.1. The constants in the previous theorem are given by

$$C_2(\rho_\lambda, \xi, \phi) = \frac{2\xi}{\phi^2}\lambda, \quad C_1(\rho_\lambda, \xi, \phi) = \frac{2\xi(1+\xi)}{\phi^2}\lambda.$$

*Proof.* Recall that $S^* := \mathrm{supp}(\theta^*)$. To lighten notation, for any vector $u$ let $u_1 := u_{S^*}, u_2 := u_{(S^*)^c}$, and also $\Delta := \widehat{\theta} - \theta^*$. Then invoking the subadditivity of $\rho_\lambda$ (this is a consequence of Condition A.1),

$$
\begin{aligned}
\rho_\lambda(\widehat{\theta}) - \rho_\lambda(\theta^*) &= \rho_\lambda(\Delta + \theta^*) - \rho_\lambda(\theta^*) \\
&= \rho_\lambda(\Delta_1 + \theta_1^*) + \rho_\lambda(\Delta_2) - \rho_\lambda(\theta_1^*) \\
&\geq -\rho_\lambda(\Delta_1) + \rho_\lambda(\Delta_2).
\end{aligned}
\tag{65}
$$

It is straightforward to derive

$$
\frac{1}{2n}\|y - Z\widehat{\theta}\|_2^2 - \frac{1}{2n}\|y - Z\theta^*\|_2^2 = \frac{1}{2n}\|Z\Delta\|^2 - \frac{1}{n}\langle w, Z\Delta \rangle.
\tag{66}
$$

Since $(w, Z) \in \mathrm{GW}_{\rho_\lambda}(\delta)$, we can invoke the GW condition with $u = \Delta$,

$$
-\frac{1}{n}\langle w, Z\Delta \rangle \geq -\frac{1}{n}|\langle w, Z\Delta \rangle| \geq -\delta \frac{1}{2n}\|Z\Delta\|^2 - \delta\rho_\lambda(\Delta).
\tag{67}
$$

It follows that

$$
\begin{aligned}
0 &\geq \frac{1}{2n}\|y - Z\widehat{\theta}\|_2^2 - \frac{1}{2n}\|y - Z\theta^*\|_2^2 + \rho_\lambda(\widehat{\theta}) - \rho_\lambda(\theta^*) \\
&\geq \frac{1}{2n}\|Z\Delta\|^2 - \frac{1}{n}\langle w, Z\Delta \rangle - \rho_\lambda(\Delta_1) + \rho_\lambda(\Delta_2) \\
&\geq \frac{1 - \delta}{2n}\|Z\Delta\|^2 - \delta\rho_\lambda(\Delta) - \rho_\lambda(\Delta_1) + \rho_\lambda(\Delta_2) \\
&= \frac{1 - \delta}{2n}\|Z\Delta\|^2 - (1 + \delta)\rho_\lambda(\Delta_1) + (1 - \delta)\rho_\lambda(\Delta_2) \\
&= (1 - \delta)\left[\frac{1}{2n}\|Z\Delta\|^2 + \rho_\lambda(\Delta_2) - \xi\rho_\lambda(\Delta_1)\right],
\end{aligned}
\tag{68}
$$

where the first inequality by optimality of $\widehat{\theta}$, the second by (66), and the third by (67). The next line follows from an an application of $\rho_\lambda(\Delta) = \rho_\lambda(\Delta_1) + \rho_\lambda(\Delta_2)$. Since $\delta < 1$ by assumption, it follows that $\rho_\lambda(\Delta_2) \leq \xi\rho_\lambda(\Delta_1)$ which implies $\Delta \in C_\rho(S^*, \xi(\delta))$.

Recalling the definition (62) of $\phi_\rho^2(Z, S^*)$, we conclude that $\frac{1}{2n}\|Z\Delta\|_2^2 \geq \frac{\phi^2}{2}\|\Delta\|_2^2$ which combined with (68), dropping $\rho_\lambda(\Delta_2)$, gives

$$
0 \geq \frac{\phi^2}{2}\|\Delta\|_2^2 - \xi\rho_\lambda(\Delta_1).
$$

Combining with the following (note $\|\Delta_1\|_0 \leq \|\theta^*\|_0$),

$$
\rho_\lambda(\Delta_1) \leq \rho_\lambda'(0+)\|\Delta_1\|_1 \leq \rho_\lambda'(0+)\|\theta^*\|_0^{1/2}\|\Delta\|_2
\tag{69}
$$

and re-arranging proves (63). For (64), since $\Delta \in C_\rho(S^*, \xi(\delta))$, we construct a set $M \subset [p]$ with $|M| = |S^*| = \|\theta^*\|_0$ such that $\Delta \in C_1(M, \xi(\delta))$. Then

$$
\begin{aligned}
\|\Delta\|_1 = \|\Delta_M\|_1 + \|\Delta_{M^c}\|_1 &\leq (1 + \xi)\|\Delta_M\|_1 \\
&\leq (1 + \xi)\|\theta^*\|_0^{1/2}\|\Delta_M\|_2 \\
&\leq \frac{2\,\xi(1 + \xi)}{\phi^2} \cdot \rho_\lambda'(0+)\|\theta^*\|_0. \qquad \square
\end{aligned}
$$

The GW condition is quantified by the constant $\delta \in (0, 1)$, and the restricted eigenvalue condition depends on the free parameter $\xi > 0$; these two are linked via the relation $\xi(\delta) = (1 + \delta)/(1 - \delta)$ and play subtle roles in the proof. A slightly modified version of this result first appeared in Zhang and Zhang [74], under different assumptions. The particular version presented here is important to derive uniform bounds for all permutations, which we discuss next.

In analogy with (23), define the following model selection exponent:

$$
\psi_\lambda(\mathbf{X}, \sigma_{\max}^2; \delta) := \inf_{0 \leq \sigma \leq \sigma_{\max}} \Phi_\lambda(\mathbf{X}, 0, \sigma^2/\delta^2).
\tag{70}
$$

1001 We often suppress the dependence on $\delta$ and write $\psi_\lambda(\mathbf{X}, \sigma_{\max}^2)$. Note that, in view of Lemma D.1,
1002 $\psi_\lambda(\mathbf{X}, \sigma_{\max}^2)$ describes the conditional probability, given $\mathbf{X}$, that $(\sigma\mathbf{w}, \mathbf{X})$ violates a GW condition,
1003 where $\mathbf{w} \sim \mathcal{N}_n(0, I_n)$ is independent of $\mathbf{X}$. More precisely,

$$
\sup_{0 \le \sigma \le \sigma_{\max}} \mathbb{P}\left[ (\sigma\mathbf{w}, \mathbf{X}) \notin \mathrm{GW}_{\rho_\lambda}^\circ(\delta) \mid \mathbf{X} \right] = \sup_{0 \le \sigma \le \sigma_{\max}} \exp[-\Phi_\lambda(\mathbf{X}, 0, \sigma^2/\delta^2)]
$$
$$
= \exp[-\psi_\lambda(\mathbf{X}, \sigma_{\max}^2)].
$$

1004 We also recall the relation

$$
\xi = \xi(\delta) = \frac{1 + \delta}{1 - \delta}. \tag{71}
$$

1005 **Proposition D.2.** *Assume that $\Sigma \succ 0$ and $\rho_\lambda$ satisfies Condition A.1. Suppose $\mathbf{X} \overset{iid}{\sim} \mathcal{N}_p(0, \Sigma)$,*
1006 *$\delta \in (0,1)$, and define $\xi$ by (71). Then there exist constants $c_0, c_1, c_2 > 0$ such that the following*
1007 *holds: If*

$$
n > c_0 \frac{\sigma_{\max}^2(1 + \xi)^2}{r_{\min}(\Sigma)} d \log p,
$$

1008 *then with probability at least $1 - c_1 \exp(-c_2 n) - p\binom{p}{d} \mathbb{E}\exp(-\psi_\lambda(\mathbf{X}, \sigma_{\max}^2; \delta))$,*

$$
\|\widehat{\beta}_j(S) - \beta_j(S)\|_2 \le C_2(\rho_\lambda, \xi, r_{\min}(\Sigma)) \cdot \|\beta_j(S)\|_0^{1/2}, \tag{72}
$$
$$
\|\widehat{\beta}_j(S) - \beta_j(S)\|_1 \le C_1(\rho_\lambda, \xi, r_{\min}(\Sigma)) \cdot \|\beta_j(S)\|_0, \tag{73}
$$

1009 *uniformly over all $j \in [p]$ and $S \subset [p]_j$.*

1010 For future reference, inspection of the proof shows that the conclusion of Proposition D.2 holds on
1011 $\mathcal{E}(\delta, \lambda) \cap \mathcal{R}(\delta)$, where

$$
\mathcal{E}(\delta, \lambda) = \left\{ \left( \widetilde{\boldsymbol{\varepsilon}}_j(S), \mathbf{X}_S \right) \in \mathrm{GW}_{\rho_\lambda}^\circ(\delta), \; \forall j \in [p], S \subset [p]_j \right\}, \tag{74}
$$
$$
\mathcal{R}(\delta) = \left\{ \phi_\rho^2(\mathbf{X}_S, m_j(S)) \ge r_{\min}(\Sigma) > 0, \; \forall j \in [p], S \subset [p]_j \right\}. \tag{75}
$$

1012 For regularizers that satisfy the lower bound in Condition A.1(c) we have the following control on
1013 the exponent $\psi_\lambda(\mathbf{X}, \sigma_{\max}^2)$:

1014 **Proposition D.3.** *Assume that $\mathbf{X} \overset{iid}{\sim} \mathcal{N}_p(0, \Sigma)$, and that $\rho_\lambda$ satisfies Condition A.1(c). Then there*
1015 *exist constants $c > 0$ and $C = C(\underline{\rho}_1, \underline{\rho}_0)$ such that for any $\delta \in (0,1)$, if*

$$
\lambda \ge C\delta^{-1}\sigma_{\max}\|\Sigma\|^{1/4}\sqrt{\frac{(d+1)\log p}{n}} \tag{76}
$$

1016 *then $\mathbb{E}\exp(-\psi_\lambda(\mathbf{X}, \sigma_{\max}^2; \delta)) \le c\exp(-\min\{2(d+1)\log p, n\})$.*

1017 The proof of Proposition D.3 follows from an argument similar to that in [74] and is omitted for
1018 brevity. In order to prove Proposition D.2, we need the following two intermediate results, providing
1019 uniform control on RE constants and GW conditions. Recall $\mathcal{E}(\delta, \lambda)$ as defined in (74).

1020 **Proposition D.4** (Uniform GW control)**.** *For any $\delta \in (0,1)$ and $\lambda > 0$,*

$$
\mathbb{P}[\mathcal{E}(\delta, \lambda)] \ge 1 - p\binom{p}{d}\mathbb{E}\exp\left[-\psi_\lambda(\mathbf{X}, \sigma_{\max}^2; \delta)\right].
$$

1021 *Proof.* Fix $\delta \in (0,1)$. By analogy with (31), for any neighbourhood $S \subset [p]_j$, let

$$
\xi_j(S) := \Phi_\lambda(\mathbf{X}_S, 0, \omega_j^2(S)/\delta^2) \ge \psi_\lambda(\mathbf{X}, \sigma_{\max}^2; \delta), \tag{77}
$$

1022 where the inequality follows from (70) and $\omega_j^2(S) \le \sigma_{\max}^2$. We follow the proof of Proposition B.5,
1023 but with $\beta_j(S)$ replaced with 0, and $\widetilde{\varepsilon}_j(S)$ replaced with $\widetilde{\varepsilon}_j(S)/\delta$. To simplify, let $\mathcal{E} = \mathcal{E}(\delta, \lambda)$,

$$
\mathcal{F}_S^j := \left\{ \left( \widetilde{\boldsymbol{\varepsilon}}_j(S), \mathbf{X}_S \right) \in \mathrm{GW}_{\rho_\lambda}^\circ(\delta) \right\},
$$

and note that $\mathcal{E} = \bigcap_{j=1}^{p} \bigcap_{S \subset [p]_j} (\mathcal{F}_S^j)^c$. According to Lemma D.1, we have

$$\mathcal{F}_S^j = \mathcal{A}\big(\widetilde{\varepsilon}_j(S)/\delta, \mathbf{X}, 0; S\big) = \mathcal{A}\big(\widetilde{\varepsilon}_j(S)/\delta, \mathbf{X}_S, 0\big)$$

where the second equality is by the same argument in the proof of Proposition B.5. Since $\widetilde{\varepsilon}_j(S)/\delta \sim \mathcal{N}\big(0, [\omega_j^2(S)/\delta^2]I_n\big)$ independent of $\mathbf{X}_S$, we conclude, using Definition A.6, that

$$\mathbb{P}\big(\mathcal{F}_S^j \mid \mathbf{X}_S\big) = \exp[-\xi_j(S)],$$

hence $\mathbb{P}(\mathcal{F}_S^j) \leq \mathbb{E}\exp[-\psi_\lambda(\mathbf{X}, \sigma_{\max}^2)], \forall S \subset [p]_j$, using the inequality in (77). The events $\mathcal{F}_S^j$ are monotonic in $S$ according to Corollary B.4. (The division of $\varepsilon_j(S)$ by $\delta$ does not change anything in that proof.) It follows that

$$\mathcal{E}^c = \bigcup_{j=1}^{p} \bigcup_{S \subset [p]_j} \mathcal{F}_S^j \subset \bigcup_{j=1}^{p} \bigcup_{T \in m_j(\Sigma)} \mathcal{F}_{M_j(T)}^j.$$

Taking the union bound, and using $|m_j(\Sigma)| \leq \binom{p}{d}$ and

$$\mathbb{P}\big[\mathcal{F}_{M_j(T)}^j\big] \leq \mathbb{E}\exp[-\psi_\lambda(\mathbf{X}, \sigma_{\max}^2)], \quad \forall T \in m_j(\Sigma),$$

finishes the proof. $\qquad\square$

**Proposition D.5** (Uniform RE control). *Assume $\mathbf{X} \overset{iid}{\sim} \mathcal{N}_p(0, \Sigma)$, $\Sigma \succ 0$, and $\rho_\lambda$ satisfies Condition A.1. There exist universal constants $c_0, c_1, c_2 > 0$, such that if*

$$n > c_0 \frac{\sigma_{\max}^2 (1+\xi)^2}{r_{\min}(\Sigma)} d(\Sigma) \log p$$

*then with probability at least $1 - c_1 \exp(-c_2 n)$,*

$$\inf_{1 \leq j \leq p} \inf_{S \subset [p]_j} \inf_{\substack{A \subset S \\ |A| \leq d}} \phi_\rho^2(\mathbf{X}_S, A; \xi) \geq r_{\min}(\Sigma).$$

The proof of this proposition follows from the results in Raskutti et al. [48] and is omitted. Recalling the definition of $\mathcal{R}(\delta)$ in (75), combined with $m_j(S) = \|\beta_j(S)\|_0 \leq d$ (cf. Definition 4.1), Proposition D.5 implies that $\mathcal{R}(\delta)$ holds with probability at least $1 - c_1 \exp(-c_2 n)$. Let us show how Proposition D.2 follows.

*Proof of Proposition D.2.* Recall the definitions of $\mathcal{E}(\delta, \lambda)$ in (74) and $\mathcal{R}(\delta)$ in (75). Propositions D.4 and D.5 guarantee that

$$\mathbb{P}\big(\mathcal{R}(\delta) \cap \mathcal{E}(\delta, \lambda)\big) \geq 1 - c_1 \exp(-c_2 n) - p\binom{p}{d}\mathbb{E}\exp(-\psi_\lambda(\mathbf{X}, \sigma_{\max}^2; \delta)).$$

Thus, it suffices to deduce (72) and (73) whenever we are on the event $\mathcal{R}(\delta) \cap \mathcal{E}(\delta, \lambda)$. The case $\beta_j(S) = 0$ follows from Proposition D.4 and Lemma D.1, and the case $\beta_j(S) \neq 0$ follows from Theorem D.1 applied to the corresponding neighbourhood regression problems. $\qquad\square$

## D.2 Some intermediate lemmas

Recall the definitions of $\mathcal{E}(\delta, \lambda)$ and $\mathcal{R}(\delta)$ in (74)–(75). We start with the following extension of GW bounds:

**Lemma D.6.** *Let $\widehat{\Delta} := \widehat{B} - \widetilde{B}(\widehat{\pi})$. On $\mathcal{E}(\delta, \lambda)$, we have*

$$\frac{1}{n}\left|\text{tr}\left(\mathbf{E}(\widehat{\pi})^T \mathbf{X}\widehat{\Delta}\right)\right| < \delta\Big[\frac{1}{2n}\|\mathbf{X}\widehat{\Delta}\|_F^2 + \rho_\lambda(\widehat{\Delta})\Big]. \tag{78}$$

*Proof.* Let $\widehat{\Delta}_j := \widehat{\beta}_j - \widetilde{\beta}_j(\widehat{\pi})$ be the $j$th column of $\widehat{\Delta}$. Then

$$\frac{1}{n}\left| \text{tr}\left(\mathbf{E}(\widehat{\pi})^T \mathbf{X}\widehat{\Delta}\right)\right| \leq \frac{1}{n}\sum_{j=1}^{p} |\langle \widetilde{\varepsilon}_j(\widehat{\pi}), \mathbf{X}\widehat{\Delta}_j\rangle|. \tag{79}$$

According to (74), on $\mathcal{E}(\delta, \lambda)$, we have $(\widetilde{\varepsilon}_j(S), \mathbf{X}_S) \in \text{GW}^\circ_{\rho_\lambda}(\delta)$ for all $S \subset [p]_j$. In particular, applying with $S = S_j(\widehat{\pi})$ and using $u = \widehat{\Delta}_j$ in the Definition D.1 of GW, we have

$$\frac{1}{n}|\langle \widetilde{\varepsilon}_j(\widehat{\pi}), \mathbf{X}\widehat{\Delta}_j\rangle| < \delta\Big[\frac{1}{2n}\|\mathbf{X}\widehat{\Delta}_j\|_2^2 + \rho_\lambda(\widehat{\Delta}_j)\Big], \quad \forall j$$

Summing over $j$ and plugging into (79) yields (78). $\qquad\qquad\square$

For any matrix $A = (a_{ij}) \in \mathbb{R}^{p\times p}$ and $S \subset [p] \times [p]$, let $A_{\langle S\rangle}$ denote the $p \times p$ matrix formed by zero-ing the elements outside of $S$, i.e.

$$(A_{\langle S\rangle})_{ij} = \begin{cases} a_{ij}, & (i,j) \in S, \\ 0, & (i,j) \notin S. \end{cases}$$

In analogy with Condition 4.1 on signal strength, let us define

$$\tau_\lambda(\alpha; \Sigma) := \inf\left\{\tau : \frac{\lambda^2}{\rho_\lambda(\tau)} \leq \frac{r_{\min}(\Sigma)}{\alpha}\right\} \tag{80}$$

where we often suppress the dependence on $\Sigma$. Note that we can write Condition 4.1 equivalently as $\tau_* \geq \tau_\lambda(a_1)$.

The next lemma is used to lower bound $\rho_\lambda(\widehat{B})$. The $\ell_0$ case is easy to prove; for completeness we prove this for $\ell_1$ and MCP.

**Lemma D.7.** *Assume that $\rho_\lambda$ satisfies Condition A.1, is right-differentiable with $\rho'_\lambda(0+) = \lambda$, and*

$$\tau_* \geq \tau_\lambda\Big(\frac{2\xi}{1-\delta_1}\Big), \quad \text{for some } \delta_1 \in (0,1) \tag{81}$$

*where $\xi = \xi(\delta)$ is defined by (71). Then, on $\mathcal{R}(\delta) \cap \mathcal{E}(\delta, \lambda)$,*

$$\rho_\lambda(\widehat{B}) \geq \delta_1\rho_\lambda(\widetilde{B}(\widehat{\pi})) + \rho_\lambda\Big((\widehat{B} - \widetilde{B}(\widehat{\pi}))_{\langle \text{supp}(\widetilde{B}(\widehat{\pi}))^c\rangle}\Big). \tag{82}$$

*Proof.* To lighten the notation, let $\Delta = \widehat{B} - \widetilde{B}(\widehat{\pi})$, $S_1 = \text{supp}(\widetilde{B}(\widehat{\pi}))$, $\Delta_1 = \Delta_{\langle S_1\rangle}$, and $\Delta_2 = \Delta_{\langle S_1^c\rangle}$. We have

$$\rho_\lambda(\Delta_1) \leq \lambda\|\Delta_1\|_1 \leq \lambda\|\widetilde{B}(\widehat{\pi})\|_0^{1/2}\|\Delta_1\|_2. \tag{83}$$

Since we are on $\mathcal{R}(\delta) \cap \mathcal{E}(\delta, \lambda)$, Proposition D.2 yields the $\ell_2$ deviation bound (72), which we use with $S = S_j(\widehat{\pi})$. Plugging into (83) and using $\|\Delta_1\|_2 \leq \|\Delta\|_2$,

$$\rho_\lambda(\Delta_1) \leq \lambda\, C_2(\rho_\lambda, \xi, r_{\min}(\Sigma)) \cdot \|\widetilde{B}(\widehat{\pi})\|_0. \tag{84}$$

Trivially, we have $\rho_\lambda(\widetilde{B}(\widehat{\pi})) \geq \rho_\lambda(\tau_*)\|\widetilde{B}(\widehat{\pi})\|_0$, so that by (84)

$$\rho_\lambda(\Delta_1) \leq \frac{\lambda\, C_2(\rho_\lambda, \xi, r_{\min}(\Sigma))}{\rho_\lambda(\tau_*)} \cdot \rho_\lambda(\widetilde{B}(\widehat{\pi})) \leq (1-\delta_1)\rho_\lambda(\widetilde{B}(\widehat{\pi})), \tag{85}$$

where the last inequality follows from (81). Finally, note that

$$\begin{aligned} \rho_\lambda(\widehat{B}) &\geq \rho_\lambda(\widetilde{B}(\widehat{\pi})) + \rho_\lambda(\Delta_2) - \rho_\lambda(\Delta_1) \\ &\geq \delta_1\rho_\lambda(\widetilde{B}(\widehat{\pi})) + \rho_\lambda(\Delta_2). \end{aligned}$$

where the first inequality is by arguments similar to those leading to (65) and the second is by (85). $\quad\square$

1068 *Remark* D.2. [62] use a slightly weaker beta-min condition in which only a constant fraction of the
1069 edges of each DAG are assumed to be sufficiently large. Lemma D.7 and the ensuing arguments carry
1070 through under such an assumption: Under Condition 3.5 in [62], we can use

$$\rho_\lambda(\widetilde{B}(\widehat{\pi})) \geq (1 - \eta_1)\rho_\lambda(\tau_*)\|\widetilde{B}(\widehat{\pi})\|_0,$$

1071 between (84) and (85) and obtain a bound similar to (82), with only the constants modified.

1072 The conclusion of Lemma D.7 is stronger than what we need in the sequel. We only use the weaker
1073 inequality $\rho_\lambda(\widehat{B}) \geq \delta_1\rho_\lambda(\widetilde{B}(\widehat{\pi}))$ implied by (82).

## D.3 A bound on the sample residuals

1075 In this section, we prove the following result, which is used in the proof of Proposition B.10:

1076 **Proposition D.8.** *Assume* $n > 4(C+1)(d+1)\log p$ *for some* $C > 0$ *and let* $\pi_0$ *be a minimum-trace*
1077 *permutation such that*

$$\frac{\rho_\lambda(\widetilde{B}(\pi_0))}{\operatorname{tr}\widetilde{\Omega}(\pi_0)} \geq \frac{1}{\delta_0}\sqrt{\frac{50(C+1)(d+1)\log p}{n}}. \tag{86}$$

1078 *Then for any* $\delta_0 > 0$, $\mathbb{P}(\mathcal{G}(\delta_0, \lambda; \pi_0)) \geq 1 - 2e^{-C(d+1)\log p}$, *i.e.*

$$\mathbb{P}\left(\frac{1}{2n}\|\mathbf{E}(\pi_0)\|_F^2 - \frac{1}{2n}\|\mathbf{E}(\widehat{\pi})\|_F^2 > \delta_0\rho_\lambda(\widetilde{B}(\pi_0))\right) \leq 2e^{-C(d+1)\log p}.$$

1079 Define two functions by

$$h_n(u) := -\frac{u^2}{n} + \frac{2u}{\sqrt{n+1}} + \frac{1}{n+1}, \quad H_n(u) := \frac{u^2}{n} + \frac{2u}{\sqrt{n}}. \tag{87}$$

1080 These functions bound the deviations in the normed residuals $\widetilde{\varepsilon}_j(\pi)$, and will be used repeatedly in
1081 the sequel. We note that

$$H_n(u) + h_n(u) \leq \frac{5u}{\sqrt{n}}, \quad u \geq n^{-1/2}. \tag{88}$$

1082 **Lemma D.9.** *Suppose* $\mathbf{w} \sim \mathcal{N}_n(0, \sigma^2 I_n)$. *Then for any* $0 < u < n/\sqrt{n+1}$,

$$\sigma^2\left(1 - h_n(u)\right) \leq \frac{1}{n}\|\mathbf{w}\|_2^2 \leq \sigma^2\left(1 + H_n(u)\right) \tag{89}$$

1083 *with probability at least* $1 - 2e^{-u^2/2}$.

1084 *Proof.* For $z \sim \mathcal{N}_n(0, I_n)$, we have the following useful bounds [see, e.g., 24, Corollary 1.2]:

$$\frac{n}{\sqrt{n+1}} \leq \mathbb{E}\|z\|_2 = \sqrt{2}\frac{\Gamma(\frac{n+1}{2})}{\Gamma(\frac{n}{2})} \leq \sqrt{n}.$$

1085 Gaussian concentration implies that for any $u > 0$, both

$$\left\{\|\mathbf{w}\|_2 \leq \sigma(n/\sqrt{n+1} - u)\right\}, \quad \text{and} \quad \left\{\|\mathbf{w}\|_2 \geq \sigma(\sqrt{n} + u)\right\}$$

1086 hold with probability at most $e^{-u^2/2}$. Thus,

$$\mathbb{P}\left(\sigma^2\left(\frac{n}{\sqrt{n+1}} - u\right)^2 \leq \|\mathbf{w}\|_2^2 \leq \sigma^2\left(\sqrt{n} + u\right)^2\right) \geq 1 - 2e^{-u^2/2}. \tag{90}$$

1087 Re-writing (90) using (87) yields the desired result. $\qquad\square$

**Lemma D.10.** *Suppose* $\mathbf{X} \overset{iid}{\sim} \mathcal{N}_p(0, \Sigma)$*. Then for any* $\pi \in \mathbb{S}_p$ *and* $0 < u < n/\sqrt{n+1}$*,*

$$\frac{1}{2} \operatorname{tr} \widetilde{\Omega}(\pi)\Big(1 - h_n(u)\Big) \leq \frac{1}{2n}\|\mathbf{E}(\pi)\|_F^2 \leq \frac{1}{2} \operatorname{tr} \widetilde{\Omega}(\pi)\Big(1 + H_n(u)\Big) \tag{91}$$

*with probability at least* $1 - 2p\binom{p}{d}e^{-u^2/2}$.

*Proof.* Note that for any $\pi \in \mathbb{S}_p$,

$$\frac{1}{2n}\|\mathbf{E}(\pi)\|_F^2 = \frac{1}{2n}\sum_{j=1}^p \|\widetilde{\varepsilon}_j(\pi)\|_2^2 = \frac{1}{2n}\sum_{j=1}^p \|\widetilde{\varepsilon}_j(S_j(\pi))\|_2^2. \tag{92}$$

Thus it suffices to bound the deviations in $\|\widetilde{\varepsilon}_j(S)\|_2$ for $S \subset [p]_j$. Consider the following events

$$\mathcal{G}_j(S) := \left\{ \frac{\omega_j^2(S)}{2}\Big(1 - h_n(u)\Big) \leq \frac{1}{2n}\|\widetilde{\varepsilon}_j(S)\|_2^2 \leq \frac{\omega_j^2(S)}{2}\Big(1 + H_n(u)\Big)\right\}$$

and let $\mathcal{G} := \bigcap_{j=1}^p \bigcap_{S \subset [p]_j} \mathcal{G}_j(S)$. By Lemma D.9, we have $\mathbb{P}(\mathcal{G}_j(S)) \geq 1 - 2e^{-u^2/2}$, for all $S \in [p]_j$. By a monotonicity argument (cf. (27)), we have $\mathcal{G} = \bigcap_{j=1}^p \bigcap_{S \in m_j(\Sigma)} \mathcal{G}_j(M_j(S))$. Applying the union bound and using (13),

$$\mathbb{P}(\mathcal{G}^c) \leq 2p\binom{p}{d}e^{-u^2/2}. \tag{93}$$

Summing the inequalities defining $\mathcal{G}_j(S_j(\pi))$, over $j$, we conclude that (91) holds on $\mathcal{G}$. The proof is complete. $\square$

Consider the (random) collection of permutations

$$\mathbb{S}_p^0 = \mathbb{S}_p^0(\delta_0; u) := \left\{ \pi \in \mathbb{S}_p : \frac{1}{2} \operatorname{tr} \widetilde{\Omega}(\pi)\Big[1 + H_n(u)\Big] \right.$$
$$\left. - \frac{1}{2} \operatorname{tr} \widetilde{\Omega}(\widehat{\pi})\Big[1 - h_n(u)\Big] \leq \delta_0 \rho_\lambda(\widetilde{B}(\pi))\right\}.$$

**Lemma D.11.** *For any* $\pi \in \mathbb{S}_p^0(\delta_0; u)$ *and* $0 < u < n/\sqrt{n+1}$*, we have*

$$\mathbb{P}\left(\frac{1}{2n}\|\mathbf{E}(\pi)\|_F^2 - \frac{1}{2n}\|\mathbf{E}(\widehat{\pi})\|_F^2 > \delta_0 \rho_\lambda(\widetilde{B}(\pi))\right) \leq 2p\binom{p}{d}e^{-u^2/2}.$$

*Proof.* Lemma D.10 implies that

$$\frac{1}{2n}\|\mathbf{E}(\pi)\|_F^2 - \frac{1}{2n}\|\mathbf{E}(\widehat{\pi})\|_F^2 \leq \frac{1}{2} \operatorname{tr} \widetilde{\Omega}(\pi)\Big[1 + H_n(u)\Big] - \frac{1}{2} \operatorname{tr} \widetilde{\Omega}(\widehat{\pi})\Big[1 - h_n(u)\Big]$$

with probability at least $1 - 2p\binom{p}{d}e^{-u^2/2}$. Since $\pi \in \mathbb{S}_p^0$, the right-side is bounded above by $\delta_0 \rho_\lambda \widetilde{B}(\pi)$ by definition, which establishes the claim. $\square$

**Lemma D.12.** $1 - h_n(u) > 0$ *for all* $u \neq 0$, $n > 0$.

*Proof.* Since $(u + \sqrt{n})^2 + 1 > 0$, re-writing this inequality yields

$$\frac{u^2}{n} + 1 > \frac{2u}{\sqrt{n}} + \frac{1}{n} > \frac{2u}{\sqrt{n+1}} + \frac{1}{n+1}$$
$$\implies 1 + \frac{u^2}{n} - \frac{2u}{\sqrt{n+1}} - \frac{1}{n+1} > 0$$

Comparing with (87) yields the claim. $\square$

*Proof of Proposition D.8.* Lemma     D.11      implies      that      for      a      choice      of $u = \sqrt{2(C+1)(d+1)\log p}$, we have

$$\mathbb{P}\left(\frac{1}{2n}\|\mathbf{E}(\pi)\|_F^2 - \frac{1}{2n}\|\mathbf{E}(\widehat{\pi})\|_F^2 > \delta_0 \rho_\lambda(\widetilde{B}(\pi))\right) \leq 2p\binom{p}{d}e^{-(C+1)(d+1)\log p}$$

$$\leq 2e^{-C(d+1)\log p}$$

for any $\pi \in \mathbb{S}_p^0$. Thus the claim will follow if we can show that $\pi_0 \in \mathbb{S}_p^0$. Note that

$$\operatorname{tr}\widetilde{\Omega}(\pi_0)\Big[1 + H_n(u)\Big] - \operatorname{tr}\widetilde{\Omega}(\widehat{\pi})\Big[1 - h_n(u)\Big]$$

$$\overset{(i)}{\leq} \operatorname{tr}\widetilde{\Omega}(\pi_0)\Big[H_n(u) + h_n(u)\Big]$$

$$\overset{(ii)}{\leq} \operatorname{tr}\widetilde{\Omega}(\pi_0)\sqrt{\frac{50(C+1)(d+1)\log p}{n}}$$

$$\overset{(iii)}{\leq} \delta_0 \rho_\lambda(\widetilde{B}(\pi_0)),$$

where (i) follows from $\operatorname{tr}\widetilde{\Omega}(\pi_0) \leq \operatorname{tr}\widetilde{\Omega}(\widehat{\pi})$ and Lemma D.12, (ii) follows by using (88) with $u = \sqrt{2(C+1)(d+1)\log p}$, and (iii) follows from assumption (86). Hence, $\pi_0 \in \mathbb{S}_p^0$ and the proof is complete. $\qquad\square$

## D.4    Proof of Proposition B.10

*Proof.* Recall the definition of $\mathcal{G}(\delta_0, \lambda; \pi)$ in (60). Fix some $\pi_0$ such that $\widetilde{B}(\pi_0) := \widetilde{B}_{\min}$ satisfies Condition 4.1 with $a_2 > 0$. Taking (arbitrarily) $C = 1$ and $\delta_0 = 10/a_2$ in Proposition D.8, we have

$$\mathbb{P}\big[\mathcal{G}(\delta_0, \lambda; \pi_0)^c\big] \leq 2e^{-(d+1)\log p}.$$

Combined with Propositions D.5 and D.4, we obtain

$$\mathbb{P}\big(\mathcal{G}(\delta_0, \lambda; \pi_0) \cap \mathcal{E}(\delta, \lambda) \cap \mathcal{R}(\delta)\big)$$

$$\geq 1 - c_1 \exp(-c_2 \min\{n, (d+1)\log p\}) - p\binom{p}{d}\mathbb{E}\exp(-\psi_\lambda(\mathbf{X}, \sigma_{\max}^2; \delta)).$$

Thus, we may assume we are on $\mathcal{G}(\delta_0, \lambda; \pi_0) \cap \mathcal{E}(\delta, \lambda) \cap \mathcal{R}(\delta)$. Since we are on $\mathcal{E}(\delta, \lambda)$, we can combine Lemma D.6 with Lemma B.9 (applied with $\pi = \pi_0$) to deduce (recall $\widehat{\Delta} := \widehat{B} - \widetilde{B}(\widehat{\pi})$)

$$\frac{1}{2n}\|\mathbf{X}\widehat{\Delta}\|_F^2 + \rho_\lambda(\widehat{B}) \leq \frac{\delta}{2n}\|\mathbf{X}\widehat{\Delta}\|_F^2 + \delta\rho_\lambda(\widehat{\Delta})$$

$$+ \frac{1}{2n}\|\mathbf{E}(\pi_0)\|_F^2 - \frac{1}{2n}\|\mathbf{E}(\widehat{\pi})\|_F^2 + \rho_\lambda(\widetilde{B}(\pi_0)).$$

Dropping the prediction loss terms (those involving $\|\mathbf{X}\widehat{\Delta}\|_F^2$), and using that we are on $\mathcal{G}(\delta_0, \lambda; \pi_0)$ to bound $\frac{1}{2n}\|\mathbf{E}(\pi_0)\|_F^2 - \frac{1}{2n}\|\mathbf{E}(\widehat{\pi})\|_F^2$, we have after rearranging,

$$\rho_\lambda(\widehat{B}) \leq (1+\delta_0)\rho_\lambda(\widetilde{B}(\pi_0)) + \delta\rho_\lambda(\widehat{B} - \widetilde{B}(\widehat{\pi})) \tag{94}$$

$$\leq (1+\delta_0)\rho_\lambda(\widetilde{B}(\pi_0)) + \delta\rho_\lambda(\widetilde{B}(\widehat{\pi})) + \delta\rho_\lambda(\widehat{B}).$$

Let $\delta_1 = 2\delta/(1-\delta)$, so that $\xi/(1-\delta_1) = (1+\delta)/(1-3\delta)$ (cf. (71)). Furthermore, since $\delta < 1/3$ by assumption, $\delta_1 < 1$, so that Lemma D.7 implies $\rho_\lambda(\widehat{B}) \geq \delta_1\rho_\lambda(\widetilde{B}(\widehat{\pi}))$ which gives (i) in (40).

Since $\rho_\lambda(\widetilde{B}(\widehat{\pi})) \leq (1/\delta_1)\rho_\lambda(\widehat{B})$, the bounds in (94) imply that

$$\rho_\lambda(\widehat{B}) \leq (1+\delta_0)\rho_\lambda(\widetilde{B}(\pi_0)) + \frac{\delta}{\delta_1}\rho_\lambda(\widehat{B}) + \delta\rho_\lambda(\widehat{B}).$$

Rearranging we get

$$\rho_\lambda(\widehat{B}) \leq \big[1 - \delta(1+\delta_1)/\delta_1\big]^{-1}(1+\delta_0)\rho_\lambda(\widetilde{B}(\pi_0)).$$

We have $[1 - \delta(1+\delta_1)/\delta_1]^{-1}(1+\delta_0) = \frac{2}{1-\delta}(1 + \frac{10}{a_2})$, using $\delta_0 = 10/a_2$ and $\delta_1 = 2\delta/(1-\delta)$ as before. This proves (ii) in (40). $\qquad\square$