[Reviews · NeurIPS 2019]

Reviewer 1



The proposed algorithm seems impractical. The problem in Equation (1) is not only nonconvex (for the MCP regularizer \rho_\lambda), but also combinatorial (for any regularizer \rho_\lambda). The combinatorial aspect comes from the constraint B \in D, where D is the set of p*p matrices representing the weighted adjacency matrix of a DAG. This problem is NP-hard. This is why [21] and [23] argue for an identifiability condition that allows recovering the node ordering in poly-time. The paper is overselling its contributions. Results in [23] do not require faithfulness, equal variance or Gaussianity. The submitted paper needs Gaussianity and equal variance. Furthermore, the submited paper is not the first to avoid faithfulness. Experimental evidence is mandatory in this particular case. Since the problem in Equation (1) is combinatorial/NP-hard, it is important to perform synthetic experiments to assess the time complexity. Minor comment: Some references labeled as Arxiv seem to actually be conference papers, for instance [21] [22] [23]. === AFTER REBUTTAL I am not satisfied with the authors response. The authors argue that reference [1] solves the problem in equation Equation (1). The authors in [1] use a heuristics, not exact optimization. Thus, my comment on practicality still stands, given that Equation (1) is NP-hard. The authors argue that their method is the first score-based method (for Gaussian, equal-variance data) to avoid faithfulness. Theoretical results in [23] do not require faithfulness, equal variance or Gaussianity. Thus, the "score-based" part does not bring any advantage. My comment about experiments was clearly due to the NP-hardness of exact optimization (not heuristics), and it still stands.

Reviewer 2



The paper is very relevant to the community. It is clearly written, though I would use fewer superlatives. I didn’t check all the proofs, but what I did check seemed solid. Regarding significance, I do believe the contribution an analysis here are excellent. Some aspects are a bit oversold, like saying “no faithfulness required”, but using a set of assumptions which cannot be cleanly untangled from such conditions. *** After authors' response *** I read the authors' response and my assessment is unchanged.

Reviewer 3



Update: The authors gave a good rebuttal, I have increased my score to 6. Original comments: In this paper, the authors considered the problem of learning directed acyclic graphs via optimizing a score. In particular, they have developed a new approach that requires O(s log p) samples to learn a DAG from the data. The proposed a approach is an optimization based approach that learns a DAG via optimizing a nonconvex scoring function. Pros. 1. The theoretical analysis of this paper is complete. In addition, the analysis techniques developed in this paper seems to be helpful to solve other related problems in structure learning and high-dimensional statistics. Cons. 1. In this paper, the authors claimed that the proposed approach is guaranteed to learn the minimum-trace DAG of the underlying distribution. However, the authors did not give any comments about the importance of learning a minimum-trace DAG in practice. It would be helpful if the authors can provide any example to show that the minimum-trace DAG is very important in applications, such as applications in biology, sociology or economics. In applications why are people interested in learning a minimum-trace DAG to represent the distribution? I understand if we assume the underlying data generating system is equivariance, then the minimum-trace DAG is the same as the DAG with equal variances, which is a particular scenario where learning the minimum-trace DAG is useful. However, the equivariance case seems too restrictive in practice. 2. Since the possible application of the minimum-trace DAG seems unclear, it is important for the author to provide real data analysis to show that minimum-trace DAG learned from the estimator (1) is really able to discover some informative results in practice. I regret that such information has not been provided in this paper. 3. It seems to me that the gap condition stated in Section 3.2 is a reasonable assumption only in the scenario where p = O(n^2). If p is beyond than that, the gap condition would become very restrictive. Is that correct? For example, if p is bigger than O(n^2), then one cannot come up with a choice of $a$ that allows gap(\Sigma) = o(1) while tolerating the average degree to grow with $n$. Hence, the gap condition is not applicable in the real high-dimensional setting where p = O(exp^n). If this is the case, perhaps the authors should state this limitation more explicitly in the paper. In conclusion, although the theoretical analysis shown in this paper seems interesting, the possible applications this estimator can be applied to is unclear.

[Author Response · NeurIPS 2019]

We thank all three reviewers for their time and feedback. Below we have done our best to respond to the major concerns.

**[R1] Score-based learning.** We'd like to emphasize that our paper focuses on *score-based learning*, and all the claims
made in the paper are specifically in regards to score-based learning. See L70–74, and in particular L73–74 where we
acknowledge that papers such as [21] and [23] achieve impressive results. These are newer methods, and do not shed
much light on score-based methods, which are very popular in practice and have a long history. Part of the purpose of
this paper is to (a) Shed light on the pros and cons of score-based methods vs. newer approaches and (b) To provide a
comprehensive, rigorous mathematical framework through which to analyze score-based methods. These contributions
have been acknowledged by R2 and R3, and we will be happy to revise the introduction and abstract accordingly to
make this distinction more clear.

**[R1] Practicality.** Please see L33–37, L43–48, and L294–301, where this is discussed explicitly. We have provided
13 total references for both exact and approximate algorithms for solving this problem. For example, this estimator
can be computed approximately using very fast coordinate descent methods for problems with 1,000s of nodes (ref
[1]). Exact computation, as acknowledged at L45–46, is of course NP-hard. Furthermore, the reduction in sample
complexity from $s^4 \log p$ to $s \log p$ is substantial (see L30–34), and raises the important question of whether or not there
exists a polynomial-time estimator that can achieve $s \log p$ sample complexity or better. Our work provides important
theoretical justification for this inquiry (in addition to the existing body of work cited above).

**[R1] Assumptions.** Please see the discussion above on score-based learning. Our analysis is the first *score-based*
*method* to avoid faithfulness, and we propose a novel proof to do this. This is important since existing proofs in
score-based learning *crucially* and *fundamentally* rely on faithfulness in order to learn a consistent structure (refs [5,
40]; see also L74–88). Regrettably, at L6–7 this is not made clear, and we will be happy to clarify this in the revision.

**[R1, R3] Real data example.** As pointed out at L43–44, our approach has been extensively studied in applications.
Please see refs [1, 26, 51, 54, 68, 76] for real data examples, including a cytometry dataset [1], gene regulatory networks
[26], S&P stock data [68], as well as real data examples from the BN repository [1, 51].

**[R2] Faithfulness.** We appreciate the reviewer's concern regarding our claims regarding faithfulness, and will be happy
to modify and clarify these claims. It is true that the connection between strong faithfulness and the beta-min condition
is not well-understood; see for example Uhler et al. (2013, AOS, p. 25): "*...a thorough analysis of the 'permutation*
*beta-min' condition and a comparison to the strong-faithfulness condition more generally is quite challenging and*
*remains an interesting open problem.*"

**[R2] Related work.** We will be happy to expand our discussions and comparisons to related work, esp. recent work
such as [21, 23, 52, 53, 73, 67], which provides an interesting contrast to the score-based method presented in our work.

**[R3] Min-trace DAGs.** We thank the reviewer for raising the important practical question of how to interpret min-trace
DAGs and what their relevance is in practice. As mentioned at L43–44, the estimator $\widehat{B}$ is very popular and used
frequently in practice (e.g. refs [1, 26, 51, 54, 68, 76]), despite our having little theoretical understanding of this. In fact,
prior to our work, it was an open question what the behavior of $\widehat{B}$ is. For example, does it converge, and if so, to what?
We note that even the former question is surprisingly tricky; please see the discussion at L211–216. One of the main
contributions of this work is to address these questions: Yes, $\widehat{B}$ converges, and we can in fact pinpoint what it converges
to. As this reviewer acknowledges, proving this is nontrivial and required developing novel analytical techniques that
can be applied more broadly. The importance of this result lies not in the fact that we might be interested in min-trace
DAGs, but perhaps that we might not be! This is the reason why we mention "caution" at L50. Whether or not one
would be interested in a min-trace (or equivariance) DAG depends on the application.

We regret that this important discussion was omitted, and will be happy to make these issues more explicit, both in the
introduction and in the conclusion, in the final version. We will also add some intuition on what min-trace means in
practice, namely the DAG that minimizes the out-of-sample predictive loss, which has obvious practical interest.

**[R3] Real data example.** Please see related discussion for R1 above.

**[R3] Gap condition.** Since the problem considered here is at least as hard as $p$ separate regression problems, requiring
$\text{gap}(\Sigma) = o(1)$ amounts to a very restrictive condition, and should not be expected in this setting. A similar argument
would apply to the $\ell_2$-estimation error as measured in Frobenius norm: We do not and should not expect $\|\widehat{B} - \widetilde{B}_{\min}\|_F$
to vanish asymptotically—instead, we expect this to scale appropriately with $p$. A more reasonable ask is that the
"average" gap per node vanishes, i.e. $\text{gap}(\Sigma) = o(p)$, which can hold in high-dimensional setting where $p = O(e^n)$, as
discussed at L204–206. In order to obtain strictly weaker results (e.g. **not** including structure consistency for $p > n$), a
gap of the same order was also assumed in [35] (Lemma 19) and [62] (Condition 5.1).

Of course, as the reviewer notes, under very strong conditions, these quantities may vanish in the high-dimensional
regime $p \to \infty$, and we are happy to add this discussion in the final version in order to clarify this point.

[Meta-Review · NeurIPS 2019]

A majority of reviewers appreciated the novel theoretical contributions of the paper, which are mainly of a statistical/information theoretical nature. A criticism raised by one of the reviewers concerns non-convexity/hardness of the considered optimization problem. Concerning this point, the rebuttal of the authors concerning the goals of the papers, the possibility of practical heuristics, and the relation to already existing literature was found convincing, and these points should be made more explicit in the final version (as proposed by the authors)